# Tidal bending of ice shelves as a mechanism for large-scale temporal variations in ice flow

Sebastian, H. R. Rosier[1] and G. Hilmar Gudmundsson[1]

[1]Department of Geopgraphy and Environmental Sciences, Northumbria University, Newcastle-upon-Tyne, NE1 8ST, UK

**Correspondence:** S. H. R. Rosier (sebastian.rosier@northumbria.ac.uk)

**Abstract.** GPS measurements reveal strong modulation of horizontal ice-shelf and ice-stream flow at a variety of tidal frequencies, most notably a fortnightly ($M_{sf}$) frequency not present in the vertical tides themselves. Current theories largely fail to explain the strength and prevalence of this signal over floating ice shelves. We show how well-known nonlinear aspects of ice rheology can give rise to widespread, long-periodic tidal modulation in ice shelf flow, generated within ice shelves themselves through tidal flexure acting at diurnal and semidiurnal frequencies. Using full-Stokes viscoelastic modelling, we show that inclusion of tidal bending within the model accounts for much of the observed tidal modulation of ice-shelf flow. Furthermore, our model shows that, in the absence of vertical tidal forcing, the mean flow of the ice shelf is reduced by almost 30 % for the geometry that we consider.

## 1   Introduction

Ocean tides are known to greatly affect the horizontal flow of both ice shelves and adjoining ice streams, even far upstream of grounding lines (GLs) (Doake et al., 2002; Brunt et al., 2010; Makinson et al., 2012; Legresy et al., 2004; King et al., 2011; Bindschadler et al., 2003b, a; Anandakrishnan et al., 2003; Alley, 1997; Gudmundsson, 2006; Marsh et al., 2013; Minchew et al., 2016; Rosier et al., 2017a). In some cases the horizontal ice flow responds at a different frequency to the tidal forcing, for example on the Rutford Ice Stream (RIS) the primary response is at a fortnightly ($M_{sf}$) frequency that is not measurable in the vertical tidal motion (Gudmundsson, 2006). More recent observations have shown that the $M_{sf}$ signal actually increases in strength on the adjoining ice shelf (Minchew et al., 2016; Rosier et al., 2017a) and also exists on isolated ice shelves which do not have large ice streams feeding into them (King et al., 2011; Gudmundsson et al., 2017).

A multitude of mechanisms have been proposed which could lead to a fortnightly modulation in ice flow: a nonlinear basal sliding law (Gudmundsson, 2007, 2011; Rosier et al., 2014), tidal perturbations in subglacial water pressure (Thompson et al., 2014; Rosier et al., 2015), grounding line migration (Rosier et al., 2014; Robel et al., 2017) and changes in the effective ice-shelf width (Minchew et al., 2016). Identifying the mechanism whereby ocean tides generate the observed tidal modulation in ice flow is important for several reasons. The amplitude of these perturbations is often a significant fraction of mean flow speed and the perturbations are widespread, impacting ice flow on a large number of ice streams and several ice shelves. Not knowing the root cause of these tidal modulations therefore implies a significant lack in our understanding of the forces controlling the large scale ice flow of the Antarctic Ice Sheet. Furthermore, there are good reasons to believe that the tidal

response is significantly affected by the rheology of ice or mechanical conditions at the base of ice streams, or possibly both in combination. Hence, once the mechanism has been fully identified, one can expect to be able to make inferences about ice rheology and/or basal conditions from observations of tidal modulations in ice flow. The Filchner-Ronne Ice Shelf (FRIS) is a particularly good natural laboratory for obtaining these insights because of the considerable tidal range, which can be as large

as 9 m (Padman et al., 2002).

Previous modelling studies have focused almost exclusively on tidal modulation of ice-stream flow (Gudmundsson, 2007, 2011; Walker et al., 2012, 2016; Thompson et al., 2014; Rosier et al., 2014, 2015; Rosier and Gudmundsson, 2016; Sergienko et al., 2009), whereas tidal modulation of the flow of ice shelves has received much less attention. This is possibly because it has often been assumed that the $M_{\mathrm{sf}}$ signal observed on ice shelves is driven by processes occurring on neighbouring ice

streams; indeed these make up the bulk of the proposed mechanisms listed above. Now that new observations show the $M_{\mathrm{sf}}$ signal strengthening downstream of GLs (Minchew et al., 2016; Rosier et al., 2017a) it has become clear that an alternative mechanism is needed which can generate this signal, independent of anything occurring on grounded ice (Minchew et al., 2016; Rosier et al., 2017a; Robel et al., 2017).

Here, we will show how the observed widespread tidal modulation in ice flow can be generated within ice shelves themselves

through tidal flexure. We begin with a description of this simple mechanism, which results directly from the well-known nonlinear aspect of the flow law of glacier ice and hence does not require an ice stream to act as a source of the observed tidal signals. Then in Sect. 3, using elastic beam theory, we derive a simple mathematical description of this mechanism that yields some insights into its importance for various ice-shelf configurations. Finally in Sect. 6, we present results from a 3-D full-Stokes viscoelastic model of a confined ice shelf, with a similar geometry to the RIS, that incorporates the new mechanism

and is capable of replicating many of the observed characteristics of the tidal response of the Ronne Ice Shelf. These results will show that this mechanism has important implications for both the time-varying and mean flow of ice shelves subjected to strong vertical ocean tides.

## 2   Flexural ice-softening mechanism

The Filchner-Ronne, Larsen and to a lesser extent Ross Ice Shelves are situated in tidally energetic regions, and thereby

subjected to large vertical motion at tidal frequencies. By far the largest tidal amplitudes are in the Weddell Sea region, particularly at the grounding line of large ice streams such as Rutford and Evans (Padman et al., 2002). In the grounding zone (here defined as a band along the grounding lines that extends several kilometers into the main shelf) the ice bends to accommodate these large vertical tidal motions. This bending generates longitudinal and shear stresses within the ice which contribute to the effective stress and are strongest near the grounding line during high and low tide. Since ice is a non-Newtonian

shear thinning fluid its effective viscosity will be altered by these tidal stresses. A schematic showing how vertical tidal motion can lead to a reduction in effective viscosity of ice shelf shear margins is shown in Fig. 1. This effect, which we will call 'flexural ice-softening', leads to an increase in ice velocity during high and low tide. We will show that this is a direct consequence of the nonlinearity of Glen's flow law.

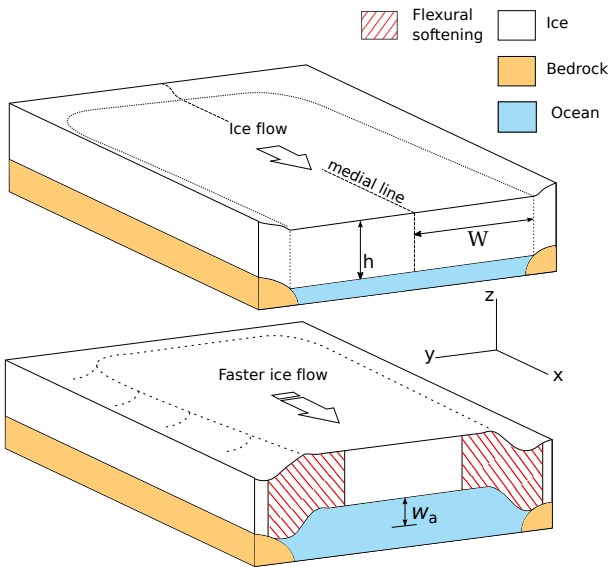

**Figure 1.** Schematic showing the flexural ice-softening mechanism for a confined shelf, together with the geometry of the problem described in Sect. 3. The top panel shows the situation with no tidal uplift and the bottom panel shows how ice flow is enhanced as ice is softened in the shear margins due to flexural stresses generated by a vertical tidal motion ($w_a$). $W$ denotes ice shelf half width and $h$ is local ice thickness.

Since it is the magnitude of stresses and not their sign that contributes to the effective viscosity, there is no difference in the flexural ice-softening effect between high and low tide. The only time that the effective viscosity of an ice shelf subjected to large tides will increase to that of an ice shelf without tides is when the vertical deflection is small, i.e. between high and low tide or during neap tides. As a consequence there are two other important repercussions for the ice-shelf flow that arise from this mechanism, aside from the direct increase in velocity at high and low tide. Firstly, the mean flow of an ice shelf is greater in the presence of large tides because, even at its slowest, it will be flowing at least as fast as an ice shelf without tides. Secondly, because the change in velocity (due to flexural ice-softening) during spring tide is larger than during neap tide, the ice-shelf flow will be modulated at an $M_{sf}$ period (provided the rheology is nonlinear, as is the case for glacier ice). Since many large ice shelves are confined on three sides by grounded ice, the bending stresses are generated along their entire length. This mechanism could therefore explain how the $M_{sf}$ signal increases in strength downstream of ice stream grounding lines, as evidenced by recent GPS and satellite observations (Minchew et al., 2016; Rosier et al., 2017a).

## 3 Analytical solution for flexural ice-softening

Elastic beam theory provides a useful starting point for evaluating the magnitude of these tidal bending stresses on an ice shelf and their impact on its effective viscosity. We start from a simple confined ice shelf whose geometry is invariant across flow (in the $y$ direction) and with a constant ice thickness gradient in the down-flow $x$ direction. The ice shelf is symmetrical about the

centerline, which is distance $W$ from the two sidewalls at $y = 0$ and $y = 2W$ (Fig. 1). For this analytical solution we assume that the portion of the ice shelf that we investigate is sufficiently far from the GL such that the only bending occurs across-flow. The situation near the main GL of a narrow confined shelf will be a complex combination of along and across-flow stresses that we shall ignore for now. Deviatoric stresses are defined as

$$\tau_{ij} = \sigma_{ij} - \delta_{ij}\sigma_{kk}/3 \tag{1}$$

where $\sigma_{ij}$ are the components of the Cauchy stress tensor, $\delta_{ij}$ is the Kronecker delta and $p = -\sigma_{kk}/3$ is the isotropic pressure. We use the comma to denote partial derivatives and the summation convention, in line with standard tensor notation.

We immediately make the simplifying assumptions (motivated by full-Stokes calculations presented below) that $\tau_{xx} = \tau_{xz} = 0$, hence $\tau_{yy} = -\tau_{zz}$, $\sigma_{zz} = -p - \tau_{yy}$ and $\sigma_{xx} = -p$. Furthermore, we assume that the only important contributions to $\tau_{yy}$ and $\tau_{yz}$ are due to tidal bending. The force balance equations in $x$ and $z$ reduce to the following form:

$$-\partial_x p + \partial_y \tau_{xy} = 0 \tag{2a}$$

$$\partial_y \tau_{yz} + \partial_z \sigma_{zz} = \rho g \tag{2b}$$

Note that in this system $\sigma_{zz}$ is not cryostatic, unlike in the shallow shelf and shallow ice approximations. We are interested in finding an expression for the across-flow variation in downstream velocity, $u(y)$, for which we need an expression for $\tau_{xy}$. As we show in appendix A, $\tau_{xy}$ is essentially independent of the tidal stresses (as well as $x$ and $z$) and can be approximated by

$$\tau_{xy} = F_d(W - y), \tag{3}$$

where $F_d = \rho g \partial_x s$.

Linear elastic beam theory gives us an expression for the elastic stresses that will arise due to tidal bending (Robin, 1958). Although strictly derived for an infinitely long ice shelf (or in the orientation of bending that we consider, infinitely wide), we show in appendix B that the equations in Robin (1958) provide a good approximation for the geometry that we are interested in. The two contributing stresses, related to the bending moment and its derivative, are the across-flow longitudinal bending stress:

$$\tau_{yy} = \frac{-6w_a \rho_w g z}{h^3 \lambda^2} e^{-\lambda y} \left[\cos(\lambda y) - \sin(\lambda y)\right] \tag{4}$$

and the across-flow shear bending stress:

$$\tau_{yz} = \frac{6\rho_w g w_a}{h^3 \lambda} e^{-\lambda y} \cos(\lambda y) \left[\frac{h^2}{4} - z^2\right], \tag{5}$$

where

$$\lambda^4 = \frac{3\rho_w g(1 - \nu^2)}{Eh^3}, \tag{6}$$

$w_a$ is the vertical tidal motion, $E$ is the Young's modulus of ice, $\nu$ is the Poisson's ratio, $h$ is local ice thickness and $\rho_w$ is the density of seawater. The vertical coordinate, $z$, is defined as the vertical distance above the neutral axis of the ice shelf, which we assume to be halfway through its thickness.

At this stage we employ a Maxwell rheological model consisting of a linear elastic spring and a nonlinear viscous dashpot, whose behaviour is modelled by Glen's law (Glen, 1955), connected in series. With this viscoelastic model the total strain is the sum of the viscous and elastic strains and the stress is equal in the two components. In this way, we can express the horizontal shear strain rate as

$$
\quad \dot{e}_{xy} = A\tau_E^{n-1}\tau_{xy} + \frac{1}{2G}\dot{\tau}_{xy} \tag{7}
$$

where

$$
G = \frac{E}{2(1+\nu)} \tag{8}
$$

and, based on the assumptions given above,

$$
\tau_E \approx \sqrt{\tau_{yy}^2 + \tau_{xy}^2 + \tau_{yz}^2}. \tag{9}
$$

Motivated both by our findings in the appendix that $\dot{\tau}_{xy} \approx 0$, and by the fact that this elastic term can only ever yield a linear response to the tidal forcing, we discard it and focus only on the nonlinear viscous response. We are concentrating on the nonlinear response because only this can explain modulation of horizontal ice-shelf flow at an $M_{\text{sf}}$ frequency, given that the $M_{\text{sf}}$ constituent is absent in the vertical tidal forcing.

By assuming that $n = 3$, we can separate the velocity into unperturbed and time-varying components. Integrating with 15 respect to $z$ and $y$ then gives the depth averaged velocity $\overline{u}$ as

$$
\overline{u}(y,t) = \frac{2A}{h}\left( \overbrace{\int_0^y h\overline{\tau_{xy}}^3 dy}^{u_0} + \overbrace{\int_0^y h\overline{\tau_{xy}} \int_b^s \tau_{yy}^2\, dzdy}^{u_{\text{long}}} + \overbrace{\int_0^y h\overline{\tau_{xy}} \int_b^s \tau_{yz}^2\, dzdy}^{u_{\text{shear}}} \right) \tag{10}
$$

where $s$ is the surface elevation, $b$ is the bed elevation and $\overline{\tau_{xy}}$ is the depth averaged shear stress. We have split this into the three components, denoted as the unperturbed ($u_0$), long(itudinal) bending stress and shear bending stress contributions to ice flow. Evaluating the integrals for each term and neglecting the overbar since everything is now depth averaged yields:

$$
\quad u_{\text{long}} = \frac{3AF_d(\rho_w g w_a)^2}{2h^4\lambda^6}\left[ e^{-\gamma}\left(1 - 2\xi + \xi\sin(\gamma) + \cos(\gamma)\left[\xi - \frac{1}{2}\right]\right) + \lambda W - \frac{1}{2}\right] \tag{11}
$$

where $\xi = \lambda W - \frac{\gamma}{2}$ and $\gamma = 2\lambda y$,

$$
u_{\text{shear}} = \frac{3AF_d(\rho_w g w_a)^2}{10h^2\lambda^4}\left[ e^{-\gamma}\left(1 - 2\xi - \xi\cos(\gamma) + \sin(\gamma)\left[\xi - \frac{1}{2}\right]\right) + 3\lambda W - 1\right] \tag{12}
$$

and

$$
u_0 = \frac{1}{2}AF_d^3\left(W^4 - (W-y)^4\right). \tag{13}
$$

The shear and across-flow longitudinal components can be combined, such that the total (time-varying) velocity $u = u_0 + \Delta u$. Along the centerline at $y = W$, the change in velocity due to tides ($\Delta u$) is

$$\Delta u = w_a^2 B, \tag{14}$$

where

$$B = \frac{3AF_d\rho_w^2 g^2}{2h^2\lambda^2}\left(e^{-\gamma}\left[\frac{1}{5} - \frac{\sin(\gamma)}{10} + \frac{1}{h^2\lambda^2} - \frac{\cos(\gamma)}{h^2\lambda^2}\right] + \frac{3\lambda W}{5} - \frac{1}{3} + \frac{W}{h^2\lambda} - \frac{1}{2h^2\lambda^2}\right) \tag{15}$$

To illustrate the consequences of a typical tidal action for the ice-shelf flow, we assume that the time-varying sea level $w_a(t)$ can be written as the sum of two cosines of amplitude $a_{M_2}$ and $a_{S_2}$ and angular frequency $\omega_{M_2}$ and $\omega_{S_2}$, i.e.

$$w_a(t) = a_{M_2}\cos(\omega_{M_2}t) + a_{S_2}\cos(\omega_{S_2}t). \tag{16}$$

These two cosines represent the principal lunar ($M_2$) and solar ($S_2$) semidiurnal tides, which dominate in the area of interest. Crucially, because the velocity is a function of tidal deflection squared, new frequencies emerge which, if we assume it takes the form of Eq. 16, expands as follows:

$$w_a^2 = \frac{a_{M_2}^2 + a_{S_2}^2}{2} + \overbrace{\frac{a_{M_2}^2}{4}\cos(2\omega_{M_2}t)}^{M_4} + \overbrace{\frac{a_{S_2}^2}{4}\cos(2\omega_{S_2}t)}^{S_4} +$$

$$\overbrace{\frac{a_{M_2}a_{S_2}}{2}\cos(\omega_{MS4}t)}^{MS_4} + \overbrace{\frac{a_{M_2}a_{S_2}}{2}\cos(\omega_{M_{sf}}t)}^{M_{sf}}, \tag{17}$$

where $\omega_{M_{sf}} = \omega_{S_2} - \omega_{M_2}$ and $\omega_{MS4} = \omega_{M_2} + \omega_{S_2}$. The four emergent frequencies that we expect to see are labelled according to their respective tidal constituent names. Depending on the relative size of the $M_2$ and $S_2$ vertical tidal forcing, different frequencies will dominate in the horizontal ice flow response. In the case of the Filchner-Ronne Ice Streams, the amplitude of the $S_2$ constituent is typically about half that of the $M_2$ constituent. As a result, the $S_4$ frequency will be much smaller than the other three. In terms of velocities, the amplitudes of the $M_{sf}$ and $MS_4$ components will be equal, and larger than the $M_4$ component as long as $a_{S_2} > a_{M_2}/2$.

Several useful results are now easily obtained with Eqs. 17 and 14, for example the amplitude of the $M_{sf}$ component in ice-shelf velocity is simply $(Ba_{M_2}a_{S_2})/2$. Integrating with time gives an expression for displacements, which are more readily measured with in-situ GPS. Once again, the amplitude of the $M_{sf}$ component in displacements in this case becomes $(Ba_{M_2}a_{S_2})/2(\omega_{S_2} - \omega_{M_2})$. Even more interesting is the result of the first term of Eq. 17, which acts to increase the time-averaged ice-shelf velocity ($u_{\mathrm{mean}}$). The size of this effect, which we call the $n_{\mathrm{shift}}$ is given by

$$n_{\mathrm{shift}} = \frac{B(a_{S_2}^2 + a_{M_2}^2)}{2}, \tag{18}$$

such that $u_{\mathrm{mean}} = u_0 + n_{\mathrm{shift}}$. Interestingly, within this framework all tidal energy at the original (vertical) semidiurnal forcing frequencies disappears (as can be seen by squaring the tidal forcing, Eqs. 16–17). In reality linear elastic effects and changes

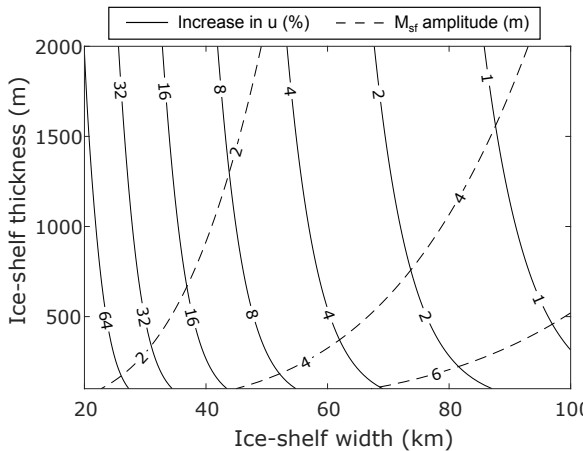

**Figure 2.** Contour plot of ice-shelf speed up due to tides, as a percent of the baseline speed, predicted by the analytical solution in Eq. 18. Speed-up is predicted along the ice shelf medial line using parameter values given in Table **??**. Also shown are contours of the amplitude of the $M_{\text{sf}}$ signal in ice-shelf displacements (dashed contours).

in damming stresses would be expected to produce some response at these frequencies and these terms are included in the 3-D model described in Section 4. Note that from Eq. 10 onwards these results have been derived under the assumption that $n = 3$. For $n = 1$ bending stresses have no impact on the ice-shelf viscosity and so the $M_{\text{sf}}$ flow-modulation and $n_{\text{shift}}$ would be identically equal to zero.

5 Using the simple set of equations outlined above we can easily explore the parameter space to see how the strength of the tidal response changes. Of particular interest is how the $n_{\text{shift}}$ leads to an increase in the mean speed of the ice shelf. In Fig. 2 we show speed-up along the ice shelf medial line (solid black contour) as a percent of the baseline speed with no tides, i.e. $u_{\text{mean}}/u_0$ (the parameters chosen are shown in Table 1). This shows that, for a given tidal amplitude, the $n_{\text{shift}}$ effect will be most strongly felt on a narrow, thin ice shelf. Conversely, the amplitude of the $M_{\text{sf}}$ signal in ice shelf displacements (dashed

10 contour) is strongest for wide, thick ice shelves. The apparent discrepancy is because, with all other parameters held constant, a wider ice shelf will flow much faster and so the increase in speed as a percent of the baseline is much less.

 Note that we use a different value of $E$ to obtain bending stresses for analytical solution than in our full-Stokes model. Using the instantaneous Young's modulus of $9\,\text{GPa}$ (suggested by laboratory experiments) would result in bending stresses that are too large. This is because ice behaves viscoelastically at tidal frequencies and $E$ is frequency dependent. This behaviour is

15 captured by our full-Stokes model but since the much simpler elastic beam model does not include this complexity then instead we treat this value as a tuning parameter and pick a value of $E$ that best matches our modelled bending stresses, which turns out to be $800\,\text{kPa}$.

**Table 1.** Choice of parameters used in Eq. 18 to produce Fig. 2.

| Parameter | Value | Unit |
|:---------:|:-----:|:----:|
| n | 3 | - |
| $a_{M_2}$ | 1 | m |
| $a_{S_2}$ | 1 | m |
| $\rho$ | 910 | $\mathrm{kg\,m^{-3}}$ |
| $\rho_w$ | 1030 | $\mathrm{kg\,m^{-3}}$ |
| g | 9.81 | $\mathrm{m\,s^{-2}}$ |
| A | $1 \times 10^{-24}$ | $\mathrm{Pa^{-3}\,s^{-1}}$ |
| $\nu$ | 0.3 | - |
| E | $8 \times 10^5$ | Pa |
| $\partial_x s$ | $5 \times 10^{-4}$ | - |

## 4  Full-Stokes Model Description

In order to explore the idea of flexural ice-softening in more detail, we undertook modelling experiments on an idealised ice stream/shelf domain using the commercial finite element software MSC.Marc, which has been used extensively in the past to explore the tidal response of ice streams (Gudmundsson, 2011; Rosier et al., 2014, 2015; Rosier and Gudmundsson, 2016).

The idealised ice stream is 28km wide (to match the approximate average width of the RIS) and consists of a 150 km floating shelf and 80 km grounded ice (Fig. 3). Although data now exists showing tidal modulation on other ice streams, the RIS lends itself well to an idealised study of this kind because of its relatively simple geometry and because its flow has remained largely unchanged over the measurement period (Gudmundsson and Jenkins, 2009). Surface and bed slopes of the ice stream and ice-shelf portions of the model are approximate averages of the slopes found on RIS, and ice thickness at the downstream limit

of the domain is 1420 m. The model is run forward in time for 60 days in order to resolve the $M_{\mathrm{sf}}$ signal. The grounding line position is fixed and cannot migrate at tidal frequencies, since our focus is only on the effects of tidal bending stresses. We investigate several test cases (Sect. 5), some of which require a slightly different model set up, which we describe in the relevant sections.

### 4.1  Field Equations

The full-Stokes solver MSC.Marc uses the finite element method in a Lagrangian frame of reference to solve the field equations:

$$\frac{D\rho}{Dt} + \rho v_{i,i} = 0, \tag{19}$$

$$\sigma_{ij,j} + f_i = 0, \tag{20}$$

$$\sigma_{ij} - \sigma_{ji} = 0, \tag{21}$$

representing conservation of mass, linear momentum and angular momentum, respectively. In the above equations, $D/Dt$ is the material time derivative, $v_i$ are the components of velocity, $\sigma_{ij}$ are the components of the stress tensor, $\rho$ is the ice density and $f_i$ are the components of the gravity force.

We use a nonlinear Maxwell viscoelastic rheology in a slightly modified form to Eq. 7, which can be written as

$$\dot{e}_{ij} = \frac{1}{2G}\overset{\triangledown}{\tau}_{ij} + A\tau_E^{n-1}\tau_{ij}, \tag{22}$$

where the full stress tensor contributes to the effective stress, i.e.

$$\tau_E = \sqrt{\tau_{ij}\tau_{ji}/2} \tag{23}$$

and the superscript $\triangledown$ denotes the upper-convected time derivative:

$$\overset{\triangledown}{\tau}_{ij} = \frac{D}{Dt}\tau_{ij} - \frac{\partial v_i}{\partial x_k}\tau_{kj} - \frac{\partial v_j}{\partial x_k}\tau_{ik} \tag{24}$$

(Christensen, 1982). We use the same rheological parameters as in Gudmundsson (2011), which are found to replicate the behaviour of the more complex Burgers model at tidal frequences, i.e. $E = 4.8\text{GPa}$ and $\nu = 0.41$, where $E = 2G(1+\nu)$ (Shames and Cozzarelli, 1997).

## 4.2 Boundary Conditions

At the downstream limit of the domain we prescribe the ice shelf stresses:

$$\sigma_{xx} = -\rho g(s-z) + \frac{\rho g h}{2}\left(1 - \frac{\rho}{\rho_w}\right) - p_b \tag{25}$$

and

$$\tau_{xz} = -\rho g z\left(\frac{\partial s}{\partial x} - \frac{1}{2}\frac{\partial h}{\partial x}\left(1 - \frac{\rho}{\rho_w}\right)\right) \tag{26}$$

where $p_b$ is a buttressing term. A value of 250kPa was chosen for $p_b$, in order to reproduce ice shelf velocities similar to those observed at the outlet of the RIS. At the upstream boundary we apply the cryostatic pressure $\sigma_{xx} = \rho g(s-z)$. At the ice surface, a stress-free boundary condition of the form $\sigma_{ij}n_j = 0$ is used, where $n_j$ is the outward unit vector normal to the surface.

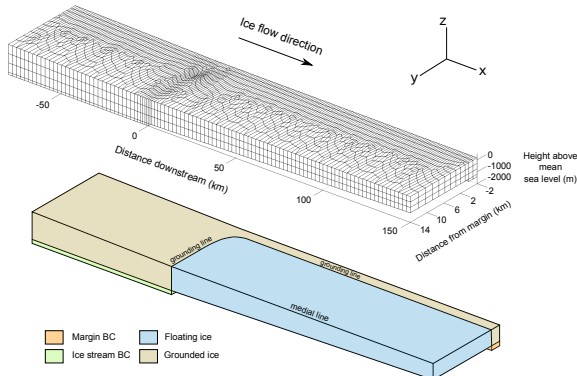

**Figure 3.** Finite element mesh used in the full-Stokes viscoelastic model (Sect. 4). Note that x and y horizontal scales have been reduced by factor 10 and 2 respectively.

The ocean pressure normal to the ice ocean interface ($p_w$) is applied as an elastic foundation (see Gudmundsson 2011 for details). This is exactly equivalent to a normal stress of:

$$p_w = -\rho_w g(z - w_a(t)) \tag{27}$$

where $z$ is the depth below sea level and $w_a(t)$ is the time varying vertical tidal motion (Sect. 5.1).

Upstream of the grounding line, along the ice-bed interface (green and orange shaded regions in Figure 3), we use a Weertman style sliding law of the form

$$u = c\tau_b^m \tag{28}$$

where $c$ is basal slipperiness, $\tau_b$ is the along-bed tangential component of the basal traction and $m$ is a stress exponent. In all of our experiments we use a nonlinear sliding law with $m = 3$. Similarly, slipperiness values beneath the ice stream are kept fixed in all experiments to a value that approximately matches the mean flow velocity of the RIS. Beneath the margin, slipperiness is made several orders of magnitude smaller to restrict ice flow in this portion of the model.

We treat one side of the model ice stream as the medial line, since the problem is symmetrical ($\partial_y h = 0$), meaning we only need to model half of the ice stream with no lateral flow as the appropriate BC. The other side is treated as a grounded sidewall with no-slip, such that $u = v = w = 0$ (referred to hereafter as the clamped BC). In one of the experiments (**n3xy**) the constraint on vertical velocity is removed, as explained in Sect. 5.

### 4.3 Discretization

The model uses 20-node isoparametric hexahedral (brick) elements with a 27-point Gaussian integration scheme. These quadratic elements allow accurate representation of stresses and strains with much fewer numbers of elements than would otherwise be needed when using linear elements. Element size varies from a maximum horizontal dimension of ∼2km to a

minimum of ∼300m around the grounding line and in the shear margins. The finite element mesh is unstructured, with a GL that curves to avoid an unnatural grounding zone corner. The ice is 3 elements thick vertically, resulting in 9 integration points through its depth. The model mesh is shown in Fig. 3. The **n3xyz** simulation (Sect. 5) was repeated with double the horizontal resolution to check if this affected results. $M_{\mathrm{sf}}$ amplitude changed by a maximum of 3%, and ice velocity by a maximum of
2.5%, and so the default resolution was deemed sufficient.

## 5   Model Experiments

We conduct three simple model experiments to investigate the effects of flexural ice-softening within our model. Model runs are named such that **n1** or **n3** denotes whether we use a linear or nonlinear ice rheolgy and **xy** or **xyz** signifies which degrees of freedom are clamped on the sidewall boundary.

**n3xyz** In the first experiment we run the model with nonlinear ice rheology and sidewalls clamped in $x$, $y$ and $z$. This is designed to simulate the 'Rutford' case whereby the margins are essentially stagnant and flexure occurs all along the GL, both where the main body of the ice stream meets the ocean and downstream of this point along the sides. In order to approximately match the observed 1m/d flow velocities of the floating portion of RIS we adjust the ice rate factor ($A$) uniformly.

**n3xy** For the second experiment we run the model as in **n3xyz** but the sidewalls downstream of the GL are not clamped vertically ($z$ direction). With this setup there is no bending along the sidewalls downstream of the GL, so flexural stresses are only generated in the grounding zone around $x = 0$. This experiment is akin to a fast flowing ice-shelf bounded by stagnant floating ice, as can be found on the floating portion of some fast flowing outlet glaciers.

**n1xyz** The third experiment uses the same setup and boundary conditions as in **n3xyz** except that ice rheology is made linear,
such that $n = 1$ in Eq. 22. This experiment is done to demonstrate the difference in response due only to changing $n$ from one to three. In this experiment therefore, the ice viscosity is not stress-dependent, such that the bending stresses do not cause a reduction in the effective viscosity of ice. As such, it is not a 'realistic' situation (since ice is known to have a nonlinear rheology) but serves to emphasise that this nonlinearity is the important one at play in our model. In order to produce sensible ice-shelf velocities, the rate factor $A$ is adjusted uniformly so that the background flow-speed
(denoted $u_{\mathrm{mean}}$ in the previous analysis) is approximately the same as the other experiments.

### 5.1   Tidal Forcing

The time-varying vertical tidal forcing is implemented as a stress acting normal to the ice shelf base (Eq. 27). For all the experiments described above the model is forced with the principal semidiurnal ($M_2$, $S_2$) and diurnal ($O_1$, $K_1$) tidal constituents, i.e. the four tidal constituents which are generally largest beneath the Ronne Ice Shelf. Their amplitudes are derived from GPS
measurements of vertical ice-shelf motion 20km downstream from RIS GL (Gudmundsson, 2006). The tidal forcing is kept intentionally simple to avoid complicating any interpretation of our full-Stokes model results.

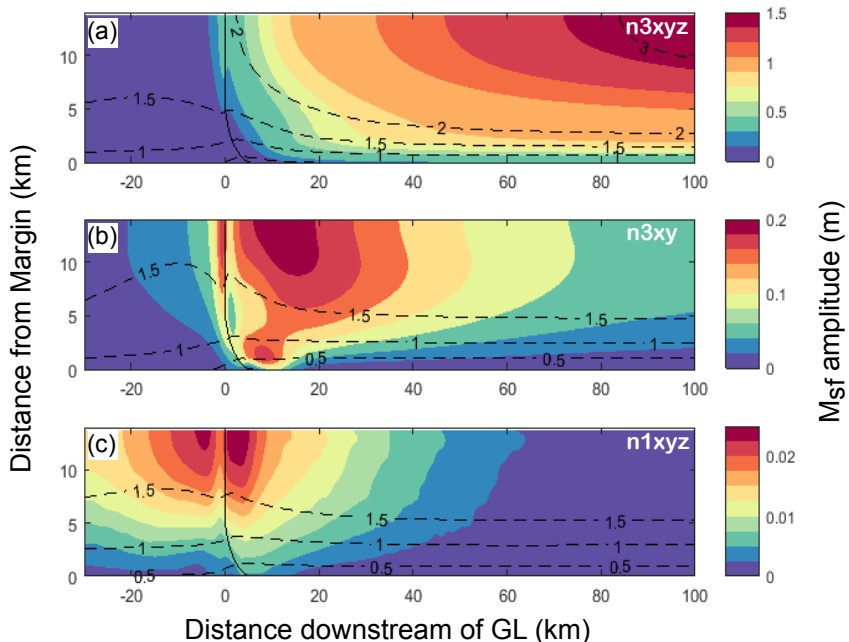

**Figure 4.** Plan view of $M_{sf}$ amplitude in horizontal surface ice displacements, calculated with the full-Stokes viscoelastic model for the three experiments described in Secion 5. Panel a shows experiment **n3xyz**, i.e. the standard case with $n = 3$ and bending all along the sidewall boundary. Panel b shows experiment **n3xy** in which the ice only bends at the $x = 0$ GL. Panel c shows the **n1xyz** experiment, for which $n = 1$ but with the same BCs as panel a. Dashed black lines are contours of downstream mean surface ice velocity and solid black lines show the GL position. Note the differences in colour scale between each panel.

## 6 Model Results

We now present results from our viscoelastic 3D full-Stokes model of an idealised ice-stream/shelf system. We begin by examining the modelled response at $M_{sf}$ frequency, since previous models do not reproduce observations of this nonlinear effect on floating ice shelves. $M_{sf}$ amplitude in horizontal surface ice displacements is shown in plan view for the three
5    experiments in Fig. 4. For the **n3xyz** experiment, which can be thought of as the typical situation for a confined ice shelf subjected to large vertical tides, $M_{sf}$ amplitude increases continuously downstream of the GL (Fig. 4a). In the across flow ($y$) direction the amplitude increases towards the medial line. Also shown are contours of ice-shelf velocity ($u$), which increase from 1 m/d upstream of the GL to more than 3 m/d on the shelf.

     In the **n3xy** experiment the only change with respect to the **n3xyz** experiment is to remove the vertical clamp BC acting
10    along the sidewall of the floating portion of the model. With this change in sidewall BC the $M_{sf}$ amplitude is similar at the $x = 0$ GL where bending stresses are still generated. Downstream of this region however the $M_{sf}$ amplitude decays rapidly to zero with distance (Fig. 4b), whereas in the **n3xyz** experiment the amplitude continues to increase with distance. Ice-velocities

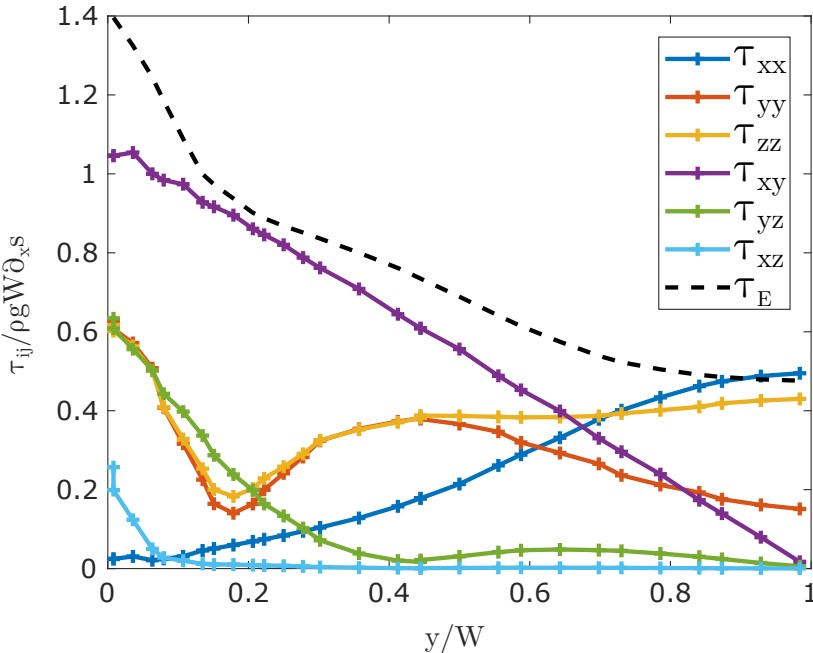

**Figure 5.** Across-flow transects of depth averaged non-dimensional stress from the full-Stokes viscoelastic model (Sect. 4) for experiment **n3xyz**. Profiles are taken 100 km downstream of the GL at high tide ($w_a = 2$ m). The stress scale is given by $\tau_{ij}/\rho g W \partial_x s$ and the length scale by $y/W$.

on the floating shelf are lower than in the **n3xyz** experiment, and across-flow shear is less pronounced, such that the ice velocity contours are further apart.

For the **n1xyz** experiment, (Fig. 4c), where the only change compared to the **n3xyz** experiment is to change the value of $n$ from one to three, the $M_{\text{sf}}$ response is even more localised to the GL region and the amplitude is close to zero.

Other tidal frequencies in the **n3xyz** experiment that emerge from the frequency doubling (Eq. 17), such as $MS_4$, show very similar spatial patterns to the $M_{\text{sf}}$ responses shown in Fig. 4a. In the **n1xyz** experiment, these frequencies are completely absent.

Running the standard **n3xyz** experiment with and without tides reveals how the mean ice-shelf flow is affected by tidal bending stresses. Averaging over the entire floating portion of the shelf, mean velocity is increased by $\sim$35% when the experiment

is run with a vertical tidal forcing equivalent to that experienced near the RIS GL, as against with no tidal forcing.

To explore the role of flexural stresses in more detail we plot across-flow profiles for each component of the deviatoric stress tensor (Fig. 5). Stresses are taken from the **n3xyz** experiment at $x = 100$ km, to avoid the 2-D bending stresses at $x = 0$, and for a positive vertical tidal deflection of 2 m. The stress is normalized by the depth-averaged horizontal shear stress at the margin $\rho g W \partial_x s$, as predicted by the analysis in Sect. 3 (for the ice-shelf surface slope in the model of $5.4 \times 10^{-4}$ the stress scale

is 67.5 kPa). Distance from the margin is normalized by the ice-shelf half-width ($W = 14$ km). Surface and bed across-flow

bending stresses ($\tau_{yy}$) are equal in amplitude but opposite in sign and so all the stresses are plotted as the depth averages of their absolute values. This is more relevant for our purposes, since it is the absolute amplitudes of these stresses, and not their signs, that impact the effective stress.

Our numerical results show that the contributions of across-flow and shear bending stresses to the effective stress, and therefore their relative impacts on effective ice viscosity, change significantly with increasing distance away from the ice-shelf margins. At the margins, both across-flow and shear bending stresses contribute about equally to the total effective stress. With increasing distance away from the margins, both bending stress terms behave as damped cosine waves (Eqs. 4 and 5), however the resulting 'waveforms' are phase shifted with respect to one another. This can be seen in Fig. 5, where $\tau_{yy}$ shows a clear minimum at a distance of $y/W \approx 0.2$ before increasing again, whereas the minimum for $\tau_{yz}$ is discernible at $y/W \approx 0.4$. As a consequence of this damped behaviour, bending stresses are largest near the grounding line but, for this geometry, have very little impact on effective viscosity along the ice shelf medial line where they have decayed to almost zero (the fact that $\tau_{yy}$ term is relatively large at the medial line is a result of ice-shelf spreading, not bending in the grounding zone). Note that, since $\lambda$ is a function of ice thickness, the location of the bending stress minima will shift as the thickness changes.

At this stage we can briefly evaluate the validity of the assumptions made in Sect. 3. The expression for the across-flow variation in $\tau_{xy}$, given by Eq. 3, varies from the value calculated by our full-Stokes model by a maximum of 5%. The assumption that $\tau_{yy} \approx -\tau_{zz}$ holds near the margin, as shown in Fig. 5 where the modelled absolute values of these two stresses are approximately equal, but begins to break down at a distance of $W/2$ where the $\tau_{xx}$ becomes increasingly large due to ice shelf spreading. Finally, the vertical shear stress ($\tau_{xz}$) is approximately zero everywhere apart from within one ice thickness of the GL, where the effects of neighbouring ice shearing vertically in the grounded margin are felt. Nevertheless, even in this region $\tau_{xz}$ contributes less than 2% of the total effective stress.

Figure **??** shows the phasing of velocity, effective stress and strain heating rates in the model shear margin relative to vertical tidal motion (vertical motion is taken along the medial line to show the undamped tidal amplitude). Strain heating rate is calculated as $\dot{e}_E \tau_E / \rho C_p$, using a specific heat capacity of 1955.4 J/K (equivalent to an ice temperature of $-20°C$, Cuffey and Paterson 2010). This shows that modelled ice velocity, effective stress and strain heating are greatest just before high and low tide, as would be expected from a viscoelastic rheology. Effective stress in the shear margin is increased by over 50% during the highest tides of the spring cycle. Strain heating rate in the shear margin is enhanced by vertical tidal motion and so this mechanism could enhance the shear heating effect which has been invoked to explain the inferred softness of Ronne Ice Shelf shear margins (Larour et al., 2005).

## 7 Discussion

The analysis of Sect. 3, together with full-Stokes viscoelastic modelling, both suggest that flexural ice-softening could play an important role in the generation of the $M_{sf}$ signal that is readily observed across the entire Ronne Ice Shelf (Rosier et al., 2017a). Flexural stresses due to vertical tidal motion can generate a fortnightly modulation in ice flow along any GL based only on the fact that ice is non-Newtonian. This mechanism is felt most strongly for a confined ice shelf, where bending occurs

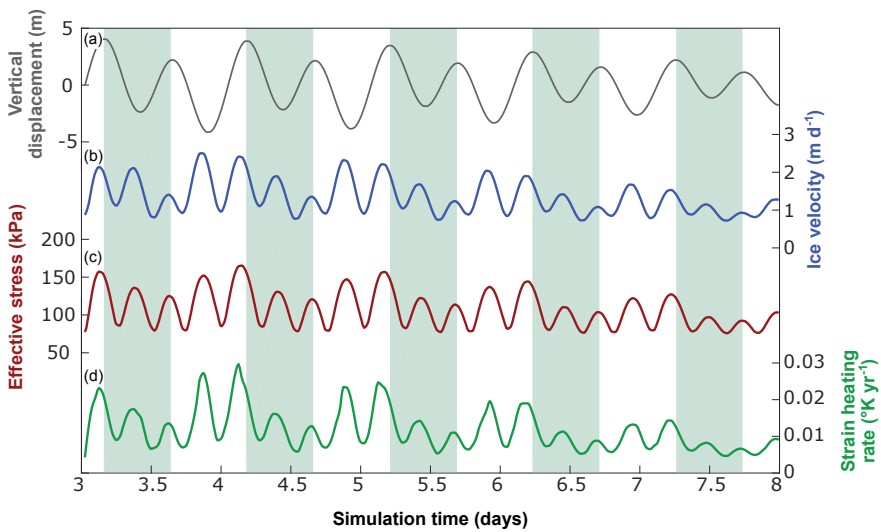

**Figure 6.** Time series of vertical ice displacement at the medial line (a), ice velocity (b), depth averaged effective stress in the shear margin (d) and depth averaged strain heating rate in the shear margin (e) from the 3D viscoelastic model. All variables are taken 100 km downstream from the main GL. The alternating blue and white shaded areas each represent one full tidal cycle, starting and ending at high tide.

in the margins along the entire length of the shelf. New observations reveal that the $M_{sf}$ signal is generally larger on ice shelves than on the adjoining ice streams, and tends to increase in amplitude in the downstream direction towards the ice front (Minchew et al., 2016; Rosier et al., 2017a). Furthermore, the $M_{sf}$ signal has now been observed to lead in phase on the ice shelf, casting some doubt on previous mechanisms that acted only on grounded ice (Minchew et al., 2016). Our modelling

work shows that flexural ice-softening can replicate this phasing and amplification of the $M_{sf}$ signal downstream of ice stream GLs. Furthermore, these tidal bending stresses will lead to a net speed-up of the ice shelf.

Two alternative mechanisms have been proposed to explain the $M_{sf}$ amplification on ice shelves, both reliant on GL migration. Minchew et al. (2016) argues that, if the sidewall GL migrates over a tidal cycle, this will lead to a change in the effective width of the ice shelf as proportionally more of it ungrounds. Observed changes in the distance between the two maxima of

lateral shear strain rate between high and low tide are interpreted as being caused by grounding line migration (Minchew et al., 2016). An alternative explanation is that flexural ice-softening in the shear margins leads to a steepening of the across-flow velocity profile at the boundary, thereby shifting the apparent margin as defined above. Calculating lateral shear strain rate 100 km downstream of the **n3xyz** simulation shows that each peak can shift by ∼500 m over a tidal cycle, leading to an apparent widening of 1 km even though there is no grounding line migration in the model. Alternative evidence of GL migration

does exist in other parts of the FRIS (Brunt et al., 2011) and this mechanism could be locally important, however, it seems unlikely that it could explain the pervasiveness of the $M_{sf}$ signal across the entire shelf, since it is so reliant on local bedrock topography.

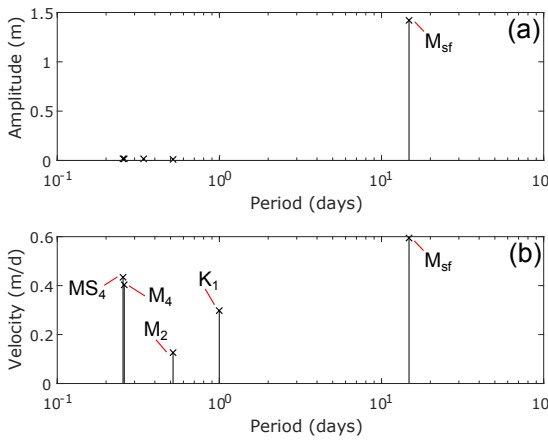

**Figure 7.** Tidal analysis of horizontal displacement (panel a) and velocity (panel b) from the full-Stokes model, taken at the medial line, 100km downstream of the GL. Notable tidal constituents are labelled with their respective names.

A previous modelling study has shown that GL migration is itself a strong nonlinearity which can generate an $M_{sf}$ response in ice flow (Rosier et al., 2014). Robel et al. (2017) explored this idea in more detail and suggested that changes in the area which an ice shelf contacts the bed (due to GL and pinning point migration) is the dominant nonlinearity on RIS leading to the observed $M_{sf}$ response. Within their framework, flexural stresses are ignored and the tidally varying ice shelf strain is a

function of competing hydrostatic and buttressing stresses. The Robel et al. (2017) model was flexible enough to allow for many of the observed aspects of the tidal modulation to be replicated. However, in the absence of a physically motivated model of GL migration, knowledge of the sub-shelf bathymetry, or even strong evidence for GL migration in the area, the extent to which this mechanism plays an important role remains an open one.

The flexural ice-softening mechanism produces a frequency doubling in the response of the ice shelf; since the marginal

ice will be softest just preceding high and low tide. This is evident in the analysis of Sect. 3, which reveals that ice shelf velocity modulation will be dominantly at $M_4$ and $MS_4$ frequencies in contrast to the $M_{sf}$ frequency which dominates the displacements. In order to check that our 3D viscoelastic model reproduces this behaviour we performed a tidal analysis on modelled displacement and velocity at the ice stream medial line, 100km downstream from the GL. Figure 7 shows the results of this tidal analysis as a frequency power spectrum, showing only constituents with a high signal to noise ratio. Surface

horizontal displacements show a dominantly $M_{sf}$ response, with almost no clear response at other frequencies (Fig. 7a). In the horizontal ice velocity (Fig. 7b) the $M_4$ and $MS_4$ frequencies emerge, with similar amplitudes to the $M_{sf}$ in agreement to Eq. 17. Other nonlinear frequencies such as $M_f$, arising from interaction of the two diurnal tidal constituents, should be present but are not resolvable with a simulation time of sixty days.

As stated above, alternative mechanisms for generating an $M_{sf}$ signal on floating ice assume that GL migration is the domi-

nant process. Ice shelf velocities from the viscoelastic model proposed by Robel et al. (2017) (using the parameters selected to

match observations on RIS) are dominated by $M_2$ and $S_2$ frequencies. Since the mechanism is nonlinear, higher frequencies such as $M_4$ and $MS_4$ are also generated, but in that model are of a lower amplitude than the semidiurnal frequencies. In order to determine which mechanism is most likely responsible for observations on the RIS, therefore, we can look at whether short-term ice shelf velocity modulation is dominantly $M_4$ and $MS_4$ or $M_2$ and $S_2$.

Most of our observations of the short-term velocity fluctuations on floating ice come from GPS units. Tidal analysis of these records is typically done on their measured displacements, rather than the much noisier velocities calculated from the time derivative of their measured position. By first fitting a tidal model to GPS measurements of horizontal ice flow downstream of the RIS, and then calculating the velocity from this smooth field, we can get a better velocity signal with which to do further analysis. A convenient measure of the importance of each tidal constituent is the percent energy (PE) (Codiga and Rear, 2004).

Tidal analysis with Utide (Codiga, 2011) of the measured horizontal ice displacements 20 km downstream of RIS GL show that the $M_{\mathrm{sf}}$ signal dominates with 87% of PE, followed by the diurnal and semidiurnal tidal constituents. Analysis of the velocites, calculated as described above, reveals that the two largest constituents are $MS_4$ and $M_4$ with 21% and 11% of PE, respectively. Based on the arguments given above, these results provide compelling evidence that the flexural ice-softening mechanism is responsible for the majority of the observed $M_{\mathrm{sf}}$ signal on the RIS.

One consequence of not including GL migration in our model is to generate artificially large stresses at the GL during high tide, where tidal stresses are acting to lift the ice from the bed but the clamped boundary condition prevents this from happening. For comparison, stresses were obtained for a simulation in which the GL was allowed to migrate, forced by a positive two metre tidal deflection. At the GL node, effective stress was 67% greater in the pinned case, but this effect is highly localised and depth averaged effective stress at the GL is only 12% greater. If bed geometry on RIS is such that the GL can

migrate a meaningful distance, our model would slightly overestimate the reduction in shear margin effective viscosity due to bending stresses at high tide. Our aim here is to investigate the flexural ice-softening mechanism in isolation and including GL migration would complicate any interpretation, particularly given the unknown bed geometry of RIS. GL migration could play a role in generating the $M_{\mathrm{sf}}$ signal observed across the Ronne Ice Shelf, depending on whether the local bed geometry permits it. That being said, both the simplicity of the flexural ice-softening mechanism, together with the ease with which it explains

many aspects of the observed tidal modulation in ice-shelf flow, suggest that it is likely to be the primary mechanism at play.

     In all our full-Stokes model experiments the $M_{\mathrm{sf}}$ signal decays rapidly upstream of the grounding line, contrary to observations which show the signal persists at least ∼80 km upstream of the Rutford, Evans and Foundation Ice Stream GLs (Gudmundsson, 2006; Minchew et al., 2016; Rosier et al., 2017a). Previous studies have proposed that a nonlinear basal sliding law could generate the $M_{\mathrm{sf}}$ signal on grounded ice (Gudmundsson, 2007, 2011; King et al., 2011; Rosier et al., 2014).

The model presented in this paper also uses a nonlinear sliding law, but when the flexural softening mechanism is absent and the nonlinear sliding law is the only mechanism at play (experiment **n1xyz**) it fails to reproduce the observed $M_{\mathrm{sf}}$ amplitude and decay length scale (Fig. 4c). Other mechanisms have been suggested which could promote propagation of this signal far upstream, for example weakened margins or tidal pressurisation of the subglacial drainage system (Thompson et al., 2014; Rosier et al., 2015). Since our focus is on the ice-shelf we do not include any of these mechanisms in this model.

The flexural softening mechanism which we have described acts in the grounding zone which may often coincide with a shear margin, a portion of the ice sheet that is complex and remains poorly understood. Shear margins are typically heavily crevassed due to the intense shear straining, making them difficult to access and instrument. These crevasses change the effective bulk properties of the ice, altering the flexural profile compared with undamaged ice (Rosier et al., 2017b). Furthermore, repeated straining will alter the ice fabric and make it highly anisotropic (Alley, 1988; Azuma, 1994). In the grounding zone, repeated tidal straining may itself alter the ice fabric, although this has never been investigated to our knowledge. Finally, lateral and tidal straining will cause strain heating (Fig. **??**d). A consequence is that ice within floating shear margins subjected to large tides may be warmer as a result of tidal flexure, although the presence of crevasses could lead to a complex depth-dependent temperature profile (Harrison et al., 1998; Perol and Rice, 2015). All of the processes described above will interact with tidal flexure and further modelling is required to evaluate their effects in detail.

Remote sensing techniques suggest that the amplitude of the $M_{sf}$ signal shows considerable spatial heterogeneity (Minchew et al., 2016). There remains some debate about the correct value for the ice rheological exponent $n$ and whether it might vary spatially (Cuffey and Paterson, 2010, and references therein), although this is often conveniently ignored in modelling studies. Since the amplitude of the $M_{sf}$ signal on the ice shelf is highly sensitive to the value of $n$, further modelling of this effect might help to provide new insights into ice rheology. For example, it might be that the observed spatial pattern and magnitude of the $M_{sf}$ effect on the shelf downstream of RIS can only be reproduced for certain choices of $n$, although it would be difficult to separate this from other factors at play. In the context of the flexural-ice softening mechanism, this heterogeneity could also arise due to variation in ice properties such as thickness, fabric, damage, etc.

## 8  Conclusions

We present results from both analytical and full-Stokes models, which show that tidal bending stresses in ice-shelf margins can give rise to large scale temporal variations in ice flow. The nonlinear rheology of ice means that, as an ice-shelf bends to accomodate vertical tidal motion, stresses generated in the grounding zone reduce the effective viscosity of ice. This leads to modulation of ice-shelf velocity at a number of frequencies, including the $M_{sf}$ frequency which is readily observed on many Antarctic ice shelves (King et al., 2011; Minchew et al., 2016; Gudmundsson et al., 2017; Rosier et al., 2017a). In addition, the nonlinear response changes the mean flow of the ice shelf when it is subjected to vertical tidal motion.

This mechanism relies only on the nonlinear rheology of ice and can explain many recent GPS and satellite observations of tidal effects on ice-shelf flow. By causing an increase in ice velocity twice during one tidal cycle, it leads to a strong frequency doubling effect which is potentially diagnosable from careful measurement of ice-shelf velocity with high temporal resolution and accuracy. Tentative analysis of GPS measurements from the floating portion of RIS suggest that these characteristic frequencies can be seen in existing data and that their relative amplitudes match those of our model.

The bending stresses investigated in this study are typically ignored and difficult to incorporate into large-scale ice-sheet models, however this work shows that these stresses have a role to play in the overall flow-regime. Full-Stokes modelling of

a tidally energetic region such as the FRIS would lead to further insights into the importance of this mechanism, its relevance for ice flow models and possibly even ice rheology.

## Appendix A: Derivation of across-flow shear stress

We start from the simplified z-momentum given in Eq. 2b, together with expressions for the bending stresses $\tau_{yy}$ and $\tau_{yz}$ (Eqs. 4 and 5 respectively). Applying the surface boundary condition $\sigma\hat{n} = 0$ we find that

$$-\partial_y s\,\tau_{yz}(s) + \sigma_{zz}(s) = 0. \tag{A1}$$

Since $\tau_{yz} = 0$ at the surface, this reveals that $\sigma_{zz}(s) = 0$.

Using this result and integrating the z-momentum (Eq. 2b) from the surface to arbitrary depth $z$ we arrive at an expression for $p(x,y,z,t)$:

$$p = \rho g(s - z) - \tau_{yy}(z) - \int_z^s \partial_y \tau_{yz}\,dz. \tag{A2}$$

Inserting this into the x-momentum of Eq. 2a gives

$$\partial_y \tau_{xy} = \rho g \partial_x s - \partial_x \tau_{yy} - \partial_x \int_z^s \partial_y \tau_{yz}\,dz, \tag{A3}$$

where

$$\partial_x \tau_{yy} = \frac{9 w_a z h^{-5/2} \partial_x h \sqrt{\rho_w E g}}{\sqrt{3(1 - \nu^2)}} e^{-\lambda y} \Big[\sin(\lambda y) + (\lambda y - 1)\cos(\lambda y)\Big], \tag{A4}$$

$$\partial_x \int_z^\hbar \partial_y \tau_{yz}\,dz = -\frac{3}{4}\rho_w g w_a \partial_x h(h - 2z) h^{-4} e^{-\lambda y}\Big(2\zeta\big[\sin(\lambda y) + \cos(\lambda y)\big] + \lambda y\big[h^2 - \zeta\big]\sin(\lambda y)\Big) \tag{A5}$$

and $\zeta = z(h + 2z)$. Note that the $x$ dependence of Eq. A2 is through the ice thickness $h$, which also appears in the expression for $\lambda$ (Eq. 6). Integrating from the surface to the bed and dividing by ice thickness yields the depth averaged across-flow gradient in horizontal shear stress:

$$\partial_y \overline{\tau_{xy}} = \rho g \partial_x s - \frac{1}{h}\int_b^s \partial_x \int_z^s \partial_y \tau_{xy}\,dz. \tag{A6}$$

With the boundary condition that $\overline{\tau_{xy}}$ is zero at the centerline, we can integrate along y to give an expression for depth averaged horizontal shear stress, which is

$$\overline{\tau_{xy}} = \rho g h \partial_x s - \frac{3\rho_w g w_a \partial_x h e^{-\lambda y} \lambda y \sin(\lambda y)}{4h}. \tag{A7}$$

It turns out that the second term on the R.H.S. of Eq. A7 is much smaller than the other two for any sensible choice in parameters and so the horizontal shear stress is balanced by the driving stress term to a very good approximation. Since the geometry along the x direction does not change with time the only temporal variation in $\tau_{xy}$ enters through the smaller second term. As such, $\dot{\tau}_{xy} \approx 0$; a curious finding given the large changes in centerline velocity but one that is borne out by examination of the stresses in our full-Stokes model (Sect. 6).

For a comparison with the idealised system of equations presented above, we take a 2-D slice through the ice shelf in the full-Stokes model (presented in Sect. 4) and look at the deviatoric stresses. We take this slice far away from the GL at $x = 0$ to avoid the additional bending stresses in this region. The lateral shear stress $\tau_{xy}$ is found to vary linearly from zero at the medial line to $\sim$70kPa at the margin and is approximately constant with depth (see also Fig 5). Maximum variation in $\tau_{xy}$ over a tidal cycle is $\sim$3%, despite the ice velocity doubling at the medial line. This matches closely with the profile predicted by Eq A7 using parameters taken from the model. The main discrepancy in stresses between the full-Stokes model and the simplified system of Eq. 2a is that modelled $\tau_{xx}$ becomes relatively large near the medial line, however since this is not the case near the margins, where most of the lateral shearing takes place, the approximation appears to not be a bad one.

## Appendix B: Analytical solution for double clamped elastic beam

Much of the work on tidal bending of floating ice is based on beam theory, specifically the analysis of elastic beams on elastic foundations first explored by Hetenyi (1946). The classical solution for bending of a floating ice tongue was first derived by Robin (1958) and has since been used extensively in studies of ice flexural process (Holdsworth, 1969, 1977; Lingle et al., 1981; Stephenson, 1984; Vaughan, 1995; Smith, 1991; Hulbe et al., 2016; Sykes et al., 2009; Rignot, 1998). We will call this set of equations the long beam model (LBM). The set of boundary conditions (BCs) chosen in the LBM are as follows:

$$\left.\begin{aligned} w &= 0 \\ w' &= 0 \end{aligned}\right\} y = 0 \qquad\qquad \left.\begin{aligned} w &= w_a \\ w' &= 0 \end{aligned}\right\} y \to \infty \tag{B1}$$

where $w(y)$ is the vertical deflection of the neutral axis and $w_a$ is the change in sea level due to tides. The assumption in Eq. B1 that ice is freely floating at the far-field boundary is valid in many circumstances, however the shelf downstream of RIS is only $\sim 30\,\text{km}$ wide and so this set of BCs might not be appropriate. A better set of BCs for a narrow ice shelf consists of a beam clamped at both ends, such that

$$\left.\begin{aligned} w &= 0 \\ w' &= 0 \end{aligned}\right\} y = 0 \qquad\qquad \left.\begin{aligned} w &= 0 \\ w' &= 0 \end{aligned}\right\} y = 2W \tag{B2}$$

Starting from the beam equation for a floating ice shelf:

$$w^{IV}(y) = -\frac{12(1-\nu^2)}{Eh^3}\rho_w g(w_a(t) - w(y)), \tag{B3}$$

subject to the BCs in Eq. B2, we arrive at the solution:

$$w(y,t) = w_a(t)\left[1 - e^{-\lambda y}\left(C_1 \sin(\lambda y) + C_2 \cos(\lambda y)\right) + e^{\lambda y}\left(C_3 \sin(\lambda y) + C_4 \cos(\lambda y)\right)\right], \tag{B4}$$

where $\lambda$ is given in Eq. 6 and the constants $C_1$ to $C_4$ are:

$$C_4 = \frac{1 - e^{2\lambda W}(\cos(2\lambda W) + \sin(2\lambda W))}{e^{4\lambda W} + 2e^{2\lambda W}\sin(2\lambda W) - 1} \tag{B5a}$$

$$C_2 = 1 + C_4 \tag{B5b}$$

$$C_3 = \frac{e^{2\lambda W}(\cos(2\lambda W) - \sin(2\lambda W)) - 1}{e^{4\lambda W} + 2e^{2\lambda W}\sin(2\lambda W) - 1} \tag{B5c}$$

$$C_1 = 1 + \frac{2\tan(2\lambda W)}{e^{4\lambda W}\tan(2\lambda W) + \tan(2\lambda W) + e^{4\lambda W} - 1} + C_4\left(\frac{e^{4\lambda W} + (3e^{4\lambda W} - 1)\tan(2\lambda W) - 1}{e^{4\lambda W} + (1 + e^{4\lambda W})\tan(2\lambda W) - 1}\right). \tag{B5d}$$

If the product $\lambda W$ is large (specifically, large in comparison to $\pi$) then the hinge zone is narrow compared to the ice shelf width. In this situation, $C_1 \approx C_2 \approx 1$ and $C_3 \approx C_4 \approx 0$, such that Eq. B4 reduces to the LBM solution (Robin, 1958). As it turns out, for the RIS where $W \approx 14$ km, this turns out to be the case and so the simpler LBM differs only very slightly from the solution given in Eq. B4. As a result, we can safely use the LBM to approximate bending stresses on the RIS.

*Competing interests.* The authors declare that they have no conflict of interest.

*Acknowledgements.* We are grateful to Rob Arthern, Brent Minchew and Teresa Kyrke-Smith for very helpful discussions and two reviewers for their constructive comments that greatly improved the quality of the manuscript. S. Rosier was funded by the UK Natural Environment Research Council large grant "Ice shelves in a warming world: Filchner Ice Shelf System" (NE/L013770/1).

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
