# Peer review of "Tidal bending of ice shelves as a mechanism for large-scale temporal variations in ice flow"

_The Cryosphere, 2017_

## Referee Comment (RC1) · VC Tsai (Referee) · 6 Nov 2017

In their manuscript, the authors describe an alternative mechanism for producing tidal modulation of ice stream motion that is dominated by fortnightly periods. Their new mechanism relies on the viscous softening caused by high deviatoric stresses induced by flexure, and has its forcing in the ice shelf rather than within grounded ice, which many other mechanisms would suggest. While the basic result is clearly described and interesting, there are a number of significant issues with the manuscript, the foremost being (1) it is not clear that the right order of magnitude of stresses can be produced if realistic bending, without an assumed fulcrum/fixed boundary condition at the ground-

ing line, is used since allowing grounding line migration significantly lowers the bending stresses; and (2) there needs to be discussion of and comparison with the recent paper of Robel et al. (Annals of Glaciology, 2017) which suggests some of the same points but has a different basic mechanism responsible. Given that the recent Robel et al. paper appears to adequately explain the observations and that the newly proposed model may overestimate its own relevance, even for a confined ice stream like Rutford, the authors need to significantly revise their manuscript to address these major issues before it will be a useful contribution. Related to this, a more direct comparison with observations seems necessary; the authors should discuss why their proposed mechanism fits observations better than other theories, which aspects of the observations point specifically to the proposed mechanism, and how general their mechanism is.

Major Comments:

1. In reality, the grounding line does not act like a fulcrum and is not fixed. Although the authors have discussed the possibility of grounding line migration somewhat, they have not discussed whether the bending stresses simulated near the grounding line might be overestimated because of the lack of migration (which alleviates the need of the grounding line to bend somewhat). Because the grounding line is assumed to be pinned ("clamped"), they cannot evaluate the possibility that asymmetries in grounding line migration may produce a strongly nonlinear ice shelf flow response (as in Robel et al. 2017, see later comment). This fixed nature of the assumed grounding line therefore seems to be a very important difference between the simulation result with reality, and must be discussed. At a minimum, the authors should describe why they expect their modeling framework to still be useful despite the simplifications.

2. It is a basic mathematical fact that a nonlinear process forced at more than one frequency will produce a response at harmonics and beats of those frequencies. The authors claim later in the paper that the flexure mechanism is the only way to produce the M4 response, but they have not proven that other nonlinear processes could not produce such a response. Indeed, Robel et al. 2017 makes this exact point in their

equations 11-13. Which brings up the next point. . .

3. There needs to be much more engagement throughout this paper with the arguments put forward by Robel et al. 2017. While we recognize that this paper was published near the time of submission of the current manuscript, the fact that the article discusses so many of the same issues, including many of the main points of the present manuscript, while also proposing a different basic mechanism related to asymmetries in contact stress from asymmetric grounding line migration, obliges the authors to discuss the Robel et al. paper and contrast their work with that work. For example, at a number of points, it is claimed that the tidal flexure mechanism is the only way to produce an increasing Msf signal in the shelf, which is also what Robel et al. 2017 claims, and the authors also claim that previous models do not reproduce observations in floating ice shelves (which is not true anymore due to the Robel work). Lines 25, 36, 155-160, 295-300, and all of the discussion and conclusions therefore need modification to be accurate and to appropriately cite the present literature.

4. (Lines 357-361 and elsewhere) What about Msf signals generated in the grounding line and then propagated downstream throughout the shelf? Wasn't this the previous explanation for the ice shelf Msf signal? Something that is not remarked upon in this paper in the temporal phasing of signals, which is important given than the Msf signal appears first in the ice shelf.

5. I agree that the elastic response can only ever yield a linear response. However, the elastic response can potentially produce a large signal at the primary tidal frequencies. The authors should at least provide an argument (in the analytic section) as to why the elastic deformation is small and so can be neglected in the analytic section.

6. It is clear from the difference between n3xyz and n3xy experiments that confinement plays an important role in producing the Msf signal at an amplitude comparable to that observed at the RIS shelf. What about unconfined shelves? Does this indicate that such shelves should have much less Msf response? What about Bindschadler and the

other FRIS ice streams? For example, does this imply that the proposed mechanism does not explain the observations of a significant Msf response at Bindschadler.

Also, it would be good to state, early on, that Rutford Ice Stream goes afloat in a trough and remains in that trough, for perhaps ∼100 km downstream of the grounding line. A map of ice velocities (like Figure 1b of Minchew 2016) would help put this in context.

7. One aspect of the Minchew 2016 observations that are not explained by this model is the along-flow variation in strain rate in the ice shelf. That study invokes a possible pinning point to produce such heterogeneity. Perhaps this should at least be remarked upon.

Minor Comments:

Line 16-17: Please rewrite for clarity: "the primary ice flow response is at a different frequency than the highest amplitude frequency of tidal forcing"

Line 26-27: Awkward sentence phrasing

Line 30: rheological behavior and the response to external forcing

Line 79-81: Would be useful if this sentence came much earlier to direct focus to the figure

Line 122: There are not separate viscous and elastic stresses. There is simply one stress which causes both viscous and elastic deformation. This sentence therefore should be rewritten to clarify. Equation 10: This is clearly only do-able in this way for n=3. Could you use a perturbation approach (or expand about \tau_xy) to solve for general n?

Line 176: How large are the linear elastic and damming stresses in comparison to the nonlinear viscous changes that you are simulating?

Line 231: It needs to be explained more clearly why the elastic modulus in the analytic calculation has to be so different from the numerical calculation. (Why does this

produce such high bending stresses in the analytic case?)

Line 240: What is an elastic foundation? Perhaps use a clearer term.

Line 245-250: Should say why you can be sure that the nonlinear sliding law isn't producing the Msf response. (I can see that it isn't because all the numerical experiments use the same sliding law.) Also, the authors should comment about what this implies about the results in Gudmundsson 2007, 2011 and other studies. . .

Figure 4: Msf amplitude in what quantity? (same comments for line 298)

Line 303-308: Should state outright here that the amplitude of Msf response is an order of magnitude less than in the n3xyz experiment.

Line 327: units of w_a?

Line 367: Would the response at M2 and S2 frequencies be larger if the elastic modulus was different? The relative difference between linear elastic and nonlinear viscous responses will be a strong function of the relative size of viscosity and elastic modulus.

Line 408: Remove "to this day"

-Victor Tsai

---

## Referee Comment (RC2) · M.P. Lüthi (Referee) · 19 Dec 2017

**Review: Rosier & Gudmundsson; tc-2017-193**

Dear colleagues,

This is an excellent paper on a very relevant topic. That being said, it could be easily extended to present some more model results, and to link it back to the problem of ice stream shear margins. Also, the problem of viscous vs. elastic stress should be carefully presented, especially in the theoretical discussion (Section 3).

My recommendation is to slightly extend the manuscript with the points mentioned below.

Sincerely, Martin Lüthi

**General comments**

The paper starts with a very well written and interesting introduction.

One point that needs attention is the intermingling of viscous and elastic stresses in the theoretical investigation (section 3). Elastic bending stresses are taken from beam theory, and then suddenly interpreted as viscous stresses. From the discussion it is not clear whether this is done because of the assumption of a Maxwell body, but then one would have to argue why elastic displacements are ignored.

It would be helpful to extend several figures, especially Figure 5 (adding $\tau_E$) and Figure 6 (adding $\tau_E$, extending it to all three experiments). Additionally, a figure showing time series of forcing, horizontal displacement, velocity, $\tau_E$ and strain heating in the shear zone would be most helpful.

One of the strong points of the paper is that no state change in the ice is required to produce the period doubling. However, there are three obvious mechanisms which should be discussed: grain size, fractural weakening and strain heating. All of these effects have been invoked to explain ice stream shear margins, so there is ample pertinent literature. It seems very likely that these processes are also active in a shelf shear margin, which is very similar to a fatigue experiment in material science. Certainly grain size will adapt to the continuous forcing, the material might suffer damaging, and strain heating (which is a model output) will warm and therefor soften the ice.

As a side-note, the authors seem to adopt (as in the recent glaciological literature) the term "full-Stokes" to mean Finite Element model, even if they don't solve the Stokes equation, but a visco-elastic extension thereof. There is no such thing as a "full-Stokes" equation, but "reduced-Stokes" solvers which ignore some terms of the Stokes equation. This is mentioned in many comments below, but should be consistently purged.

**Specific comments**

- 29 awkward end of sentence.
- 39 GL has not been defined (meaning grounding line).
- 40 I think this should be "independent"
- 62 To my understanding the effect should be greatest during periods of highest flexuring rates, i.e. during rising and falling tides. The reason is that viscous stresses are created by viscous deformation.

Consequently, also here should be "rising" and "falling" tide (not high and low).

here again the displacement is alluded to, but that is something elasticity is concerned about.

"equally as fast": check the usage of English

"Cauchy" upper case (it's a name)

There is no "Stokes flow" in these equations, that's just the force balance. (For Stokes flow you need the rheology).

"strain of the two components".

This equation is based on Glen's flow law, which was never introduced.

where does the factor 2 in the first term come from? Using standard Glen's flow law its just $\dot{\epsilon}_{xz} = A\tau_E^{n-1}\tau_{xy}$. Maybe you use non-standard definitions (as compared to glaciology textbooks), but general definitions should be given.

usually $\tau_E^2 = \frac{1}{2}\left(\tau_{xy}^2 + \ldots\right)$

How is $M_{sf}$ absent? Tidal forcing is mainly vertical.

I don't understand how horizontal integration yields a vertical average.

Eq 10 explain that $s$ and $b$ refer to surface and bed. Also I think that units don't match since you integrate a stress divided by $h$ (which the bar seems to indicate), to different powers.

Eq 17 in the overbrace of the third term: $S_4$ (currently no index)

and Eq. (10): somehow viscous deformation rates are obtained from elastic stresses, since $\lambda$ and the $\tau$'s are derived from elastic constants (Eq. (6)). IIUC this should be obtained from the viscous analog stresses with $\mu = 0.5$ (for incompressibility).

It would be helpful mentioning why (integration, step from (16) to (17)).

Absolute speedup would also be interesting. IIUC, the relative speedup for small shelves is not due to $n_{shift}$, but due to an increase of $u_0$.

should this be "Table 1"?

"dependant" should be "dependent".

Again, viscous stresses are not the same elastic stresses. The cap on the value of $E$ seems to be a hint in this direction: viscous stresses cannot be much higher for ice rheology since strain-rate softening limits the maximum sustainable stress.

Was the model surface geometry prescribed, or free to evolve to a consistent state?

Do you mean "Eulerian frame of reference" since total derivatives (Jaumann) appear below? These wouldn't be needed in a Lagrangian frame.

give a reference to the Equation how to calculate $G$ in Eq. (22) from $E$ and $\mu$.

Eq 25 why is $\rho_w g w_a(t)$ not included here?

Fig 3 It might be important to highlight that the grounding line is not just upstream, but also sideways of the model.

Which physical process generates this buttressing stress?

and Eqs. (25) and (26): use consistently $\rho_i$, or drop the index "i" consistently.

I have no idea what an "elastic foundation" is. It looks like you just apply a normal stress on all faces in contact with the ocean, or at least this is what it should be. "Elastic" makes no sense here.

If this corresponds to the green "till" area in Figure 3, please say so.

How is this boundary condition implemented? Is a Dirichlet (i.e. velocity) condition prescribed, the magnitude of which is calculated from the basal stress of the last time step solution?

simply call this section "Discretization"

Leave away this sentence, we typically also use HEX20 or even better HEX27, so this is standard.

The model mesh appears extremely coarse for the task at hand. Especially the horizontal discretization in a shear zone should be considerably smaller to resolve the stress concentration there, and the element layer on the sides should be bigger than 2 elements. Or at least some model experiments with double mesh density should be used to show that the chosen resolution is sufficient to resolve the relevant spatial scales.

Why was a mesh with such a complicated structure chosen for a block, where a structured mesh, possibly refined in the areas of interest, would have sufficed?

Is the mesh moving? If not, how big are the errors in transient stresses?

ff.: better explain what n1, n3 means, and leave away the "denoted n3xyz" and similar.

space missing in $x, y, z$.

Fig 4 It is somewhat confusing to have the horizontal axis reversed as compared to Figures 1 and 3. Better mirror those.

Also indicate what is shown with these contours. Are these velocity amplitudes? Or vertical displacement amplitudes at the surface, or the base of the ice? Or horizontal displacements?

Additionally, it would help saying that these are map-plane views (I think), of maybe the surface, or average, quantities.

Instead of n=1, n=3, the panels should be labeled with the codes of the model experiments e.g. n1xyz.

Velocity contours could also be shown for lower velocities, e.g. 0.1, 0.5, 1., 2., such that panels b and c show something interesting now this mysterious term is called the "ocean foundation BC", where it was "elastic" in line 240. Just call this the normal stress. Also parenthesis is missing.

So why FRIS, if everything else, including GPS, is from RIS?

Don't call this "full-Stokes". Either you solve the Stokes equations, or you don't. But here you claim to solve with a viscoelastic rheology, which is again something quite different.

see 294, and please don't perpetuate this stupid "full-Stokes" thing.

Please specify which displacements these are: horizontal or vertical? One would expect to see velocities, given the theoretical section and the model description.

"medial line": better write "ice shelf center line"

better "For experiment n1xyz …"

This scaling makes the comparison with Figure 4 difficult. At least you should indicate the grounded portion on the horizontal scale, this might be estimated to be 0.1? Or even less, since the crosses are the stresses extracted at element centers, or element face centers (actually maybe integration points, unless projected to the nodes)?

Since you have symbols, please use those in the descriptions.

units of $w_a = 2\,m$. But this was already said in line 321.

Don't call yy "longitudinal", this is plain confusing. It is the cross-flow component.

where is $\pi/4\lambda$ in the figure? Please help the reader in the figure, or improve the description.

Probably first tell the reader that $\tau_{xy}$ is the boring component holding back the weight of the ice, that you would get irrespective of rheology.

Also show, and discuss, $\tau_E$, this is the main point of the paper.

again, no full-Stokes. And as said just above, this part is boring since you get that with any rheology, it's just force balance.

Fig 5 Indicate the grounded part with a colored bar along the horizontal axis, or a vertical dotted line, or similar.

 Also plot the effective stress, since this is the quantity that is crucial for the whole argument.

This section should go to the discussion.

Why is $\tau_{xx}$ increasing downstream? Is this already the effect of the end of the domain, or the standard ice shelf extensional stress?

again, why full-Stokes? This is visco-elastic!

No, you have not shown that, at least not in the paper. Why aren't any plots of the velocity-spectrum, together with the forcing, shown. Or at least curves of the time-variation of forcing and response.

 After reading on I found Figure 6 which sort of shows this. So why is this not presented in the "Results" section, such that it could be meaningfully discussed?

I sincerely doubt this claim when using a spatial resolution of only $300\,m$. This effect might be quite important, but will be smoothed out by the horizontal and vertical approximation functions.

This period-doubling has been nicely shown in the theoretical part, although with some doubts concerning the problem with viscous/elastic stresses. Now one would like to see this frequency-doubling also in the FE model results. So it would be very nice to have a plot with forcing and response at different points on the domain.

Maybe the argument is that the stresses are highest at high tide, and therefore viscous deformation rates. But since everything is transient, and you have a visco-elastic membrane that is bent up and down, there must be location dependent delays.

Why at high and low tide? Vertical velocities are highest during rising and lowering. You could/should investigate this claim by just extracting the second invariant from your transient model runs.

"medial line"

Fig 6 It is not clear what is shown in this figure. Is this the FFT (frequency spectrum) of the model response? What was the strength of the forcing? It would be very helpful to also show the time series.

Panel (b) should be labeled with (m/d), as it shows velocities.

Panel (c) (missing) should show the effective stress in the shear margin.

This figure would also be a good place to compare the results from the three model runs, maybe as colored bars.

"...than the run time of 60 days".

I think this is "vice versa" (good old Latin).

Again, no "full-Stokes".

Add some discussion of additional, state-changing processes here (see general comment).

---

## Author Comment (AC2) · 4 Jan 2018

Sebastian H. R. Rosier and G. Hilmar Gudmundsson

sebsie46@bas.ac.uk

We thank Martin Lüthi for his helpful comments, our responses to each of the main points (in bold) are included below (in italics). All of the minor comments will be addressed in a revised version of the manuscript.

**One point that needs attention is the intermingling of viscous and elastic stresses in the theoretical investigation (section 3). Elastic bending stresses are taken from beam theory, and then suddenly interpreted as viscous stresses. From the discussion it is not clear whether this is done because of the assumption of a Maxwell body, but then one would have to argue why elastic displace-**

**ments are ignored.**

*In section 3 we take bending stresses from elastic beam theory and use them in a Maxwell rheological model. With a Maxwell rheology, the stresses in the elastic and viscous components are equal and so this is a reasonable approach. Next, we throw away the elastic deformational component and we justify this for two reasons. Firstly, as we show in appendix A, the time derivative of the across flow shear stress is negligible, and so this elastic term will become very small. Secondly, we are primarily interested in how an ice shelf can generate a nonlinear Msf response to a tidal forcing and this elastic term will only directly yield a linear response. These points are made in the text but could be clearer and so we will emphasise them in the revised manuscript.*

**It would be helpful to extend several figures, especially Figure 5 (adding $\tau$E) and Figure 6 (adding $\tau$E, extending it to all three experiments). Additionally, a figure showing time series of forcing, horizontal displacement, velocity, $\tau$E and strain heating in the shear zone would be most helpful.**

*We will explore ways to include adding the effective stress into the figures without adding too much clutter, we agree that this would be helpful for the reader. However figure 6 is a periodogram and it is unclear how this could be incorporated. A figure showing the time series of forcing etc. will also be added.*

**One of the strong points of the paper is that no state change in the ice is required to produce the period doubling. However, there are three obvious mechanisms which should be discussed: grain size, fractural weakening and strain heating. All of these effects have been invoked to explain ice stream shear margins, so there is ample pertinent literature. It seems very likely that these processes are also active in a shelf shear margin, which is very similar to a fatigue experiment in material science. Certainly grain size will adapt to the continuous forcing, the material might su er damaging, and strain heating (which is a model output) will warm and therefor soften the ice.**

*We agree that a discussion on these other shear margin processes will greatly add to our paper and will add one to the revised manuscript.*

**As a side-note, the authors seem to adopt (as in the recent glaciological literature) the term "full-Stokes" to mean Finite Element model, even if they don't solve the Stokes equation, but a visco-elastic extension thereof. There is no such thing as a "full-Stokes" equation, but "reduced-Stokes" solvers which ignore some terms of the Stokes equation. This is mentioned in many comments below, but should be consistently purged.**

*In our numerical treatment we include all terms of the momentum equations, apart from the acceleration terms. In the glaciological literature the resulting form of the momentum equations is commonly referred to as the 'full Stokes' equations, and to remain consistent with previous work we use this terminology (it is certainly not intended as a substitute for Finite Element model). It is important that a reader understands that our model includes all the equilibrium stress balance terms, as against other commonly used approximations in glaciology such as the SSA/SIA. The term 'reduced-Stokes' is likely to cause confusion in this regard, but we are happy to go with the editor's recommendation.*

---

## Author Response (AR1)

We thank both Martin Lüthi and Victor Tsai for their helpful comments, which have helped greatly improve our manuscript. Our responses to each of their points (in bold) are included below (in italics).

**Reply to Martin Lüthi**

**One point that needs attention is the intermingling of viscous and elastic stresses in the theoretical investigation (section 3). Elastic bending stresses are taken from beam theory, and then suddenly interpreted as viscous stresses. From the discussion it is not clear whether this is done because of the assumption of a Maxwell body, but then one would have to argue why elastic displacements are ignored.**

*In section 3 we take bending stresses from elastic beam theory and use them in a Maxwell rheological model. With a Maxwell rheology, the stresses in the elastic and viscous components are equal and so this is a reasonable approach. Next, we throw away the elastic deformational component and we justify this for two reasons. Firstly, as we show in appendix A, the time derivative of the across flow shear stress is negligible, and so this elastic term will become very small. Secondly, we are primarily interested in how an ice shelf can generate a nonlinear Msf response to a tidal forcing and this elastic term will only directly yield a linear response. These points are made in the text but could be clearer and so we have emphasised them in the revised manuscript.*

**It would be helpful to extend several figures, especially Figure 5 (adding τE) and Figure 6 (adding τE, extending it to all three experiments). Additionally, a figure showing time series of forcing, horizontal displacement, velocity, $τ_E$ and strain heating in the shear zone would be most helpful.**

*Figure 6 is a periodogram and it is unclear how tE could be incorporated, but we agree that a figure showing the time series of forcing etc. would be very useful and this has been added.*

**One of the strong points of the paper is that no state change in the ice is required to produce the period doubling. However, there are three obvious mechanisms which should be discussed: grain size, fractural weakening and strain heating. All of these effects have been invoked to explain ice stream shear margins, so there is ample pertinent literature.**
**It seems very likely that these processes are also active in a shelf shear margin, which is very similar to a fatigue experiment in material science. Certainly grain size will adapt to the continuous forcing, the material might suffer damaging, and strain heating (which is a model output) will warm and therefor soften the ice.**

*We agree that a discussion on these other shear margin processes will greatly add to our paper and we have added one to the discussions.*

**As a side-note, the authors seem to adopt (as in the recent glaciological literature) the term "full-Stokes" to mean Finite Element model, even if they don't solve the Stokes equation, but a visco-elastic extension thereof. There is no such thing as a "full-Stokes" equation, but "reduced-Stokes" solvers which ignore some terms of the Stokes equation.**
**This is mentioned in many comments below, but should be consistently purged.**

*In our numerical treatment we include all terms of the momentum equations, apart from the acceleration terms. In the glaciological literature the resulting form of the momentum equations is commonly referred to as the 'full Stokes' equations, and to remain consistent*

with previous work we use this terminology (it is certainly not intended as a substitute for Finite Element model). It is important that a reader understands that our model includes all the equilibrium stress balance terms, as against other commonly used approximations in glaciology such as the SSA/SIA. The term 'reduced-Stokes' is likely to cause confusion in this regard, but we are happy to go with the editor's recommendation and change our terminology accordingly.

**Specific comments**

**29 awkward end of sentence.**

*Changed the wording.*

**39 GL has not been defined (meaning grounding line).**

*Added a definition slightly before this point at the first instance of grounding line in the introduction*

**40 I think this should be "independent"**

*Fixed*

**62 To my understanding the effect should be greatest during periods of highest flexuring rates, i.e. during rising and falling tides. The reason is that viscous stresses are created by viscous deformation.**

*The reviewer is correct that simply saying the effect is greatest at high and low tide is an oversimplification, however since ice is viscoelastic at tidal frequencies the greatest softening effect will be somewhere between the time of maximum flexural rate and high tide.*
*For an oscillatory stress (i.e. tidal pressure) acting on a Maxwell material, strain will oscillate at the same frequency but with a phase lag. For an ideal elastic material the lag would be zero, whereas for a viscous material the phase lag would be pi/2. Thus, maximum strain will lag slightly after high and low tide, and so strain rate (i.e. velocity) will be greatest slightly before high and low tide. This can be seen in our new figure and we have reworded the relevant sections to make this point clear.*

**65 Consequently, also here should be "rising" and "falling" tide (not high and low).**

*See response above.*

**69 here again the displacement is alluded to, but that is something elasticity is concerned about.**

*The point being made here is that during neap tide (when vertical displacements are very small) the effect on ice shelf flow will be negligible. This is fundamental to the mechanism of the paper, since that is what leads to a fortnightly Msf signal in ice shelf displacements.*

**73 "equally as fast": check the usage of English**

*Changed to: "at least as fast"*

**93 "Cauchy" upper case (it's a name)**

*Fixed*

**98 There is no "Stokes flow" in these equations, that's just the force balance. (For Stokes flow you need the rheology).**

*Changed to "force balance"*

**123 "strain of the two components".**

*Fixed*

**124 This equation is based on Glen's flow law, which was never introduced.**

*Explained that this viscoelastic equation is based on Glen's flow law and added a reference.*

**124 where does the factor 2 in the first term come from? Using standard Glen's flow law it's just $\varepsilon_{xy} = A\tau_E^{n-1}\tau_{xy}$ . Maybe you use non-standard definitions (as compared to glaciology textbooks), but general definitions should be given.**

*This factor 2 somehow crept in too early and should not be in Eq. 7, however the rest of the Eqs. are correct since, under our assumptions, $\dot\varepsilon_{xy} = \frac{1}{2}\frac{du}{dy}$, Eq. 7 has been corrected.*

**128 usually $\tau_E^2 = \frac{1}{2}(\tau_{xy}^2 + \cdots)$**

*Since, in the simple case that we outline, $\tau_{xx}^2$ = 0 and $\tau_{yy}^2 = \tau_{zz}^2$ , the standard definition for effective stress, i.e.*

$$\tau_E^2 = \frac{1}{2}\left(\tau_{xx}^2 + \tau_{yy}^2 + \tau_{zz}^2\right) + \tau_{xz}^2 + \tau_{xy}^2 + \tau_{yz}^2$$

*Becomes*

$$\tau_E^2 = \frac{1}{2}\left(2\tau_{yy}^2\right) + \tau_{xz}^2 + \tau_{xy}^2$$

*And hence leads to equation 9.*

**133 How is Msf absent? Tidal forcing is mainly vertical.**

*The Msf frequency is totally absent from the vertical tidal motion, hence why a linear process could not produce it in the horizontal ice flow. We have reworded a sentence in the introduction to emphasise that the Msf frequency is not measurable in the vertical tidal motion.*

**135 I don't understand how horizontal integration yields a vertical average.**

*Reworded this to say "integrating with respect to z and y"*

**Eq 10 explain that s and b refer to surface and bed. Also I think that units don't match since you integrate a stress divided by h (which the bar seems to indicate), to different powers.**

*Added definitions of s and b, also corrected the typo in Eq. 10.*

**Eq 17 in the overbrace of the third term: S4 (currently no index)**

*Fixed*

**140 and Eq. (10): somehow viscous deformation rates are obtained from elastic stresses, since λ and the τ 's are derived from elastic constants (Eq. (6)). IIUC this should be obtained from the viscous analog stresses with μ = 0.5 (for incompressibility).**

*As explained above, using a Maxwell rheology the stress is equal in the viscous and elastic components.*

**175 It would be helpful mentioning why (integration, step from (16) to (17)).**

*Added this explanation*

**184 Absolute speedup would also be interesting. IIUC, the relative speedup for small shelves is not due to nshift, but due to an increase of u0.**

*We intentionally talk about relative rather than absolute speedup because, since all other parameters are kept the same (i.e. we don't change ice viscosity), the majority of absolute speedup occurs for larger shelves because they are less buttressed. Since we want to show how the ice shelf speeds up only due to the n-shift effect, we show the relative speedup compared to the baseline ice shelf velocity with no tides.*

**185 should this be "Table 1"?**

*Fixed*

**191 "dependant" should be "dependent".**

*Fixed*

**192 Again, viscous stresses are not the same elastic stresses. The cap on the value of E seems to be a hint in this direction: viscous stresses cannot be much higher for ice rheology since strain-rate softening limits the maximum sustainable stress.**

*In the Maxwell model, viscous stresses are exactly equal to elastic stresses.*

**210 Was the model surface geometry prescribed, or free to evolve to a consistent state?**

*We use a stress-free boundary condition at the ice surface, this BC is now given formally in the relevant section of the model description.*

**213 Do you mean "Eulerian frame of reference" since total derivatives (Jaumann) appear below? These wouldn't be needed in a Lagrangian frame.**

*We are slightly confused by this question, firstly total (material derivatives) are different from Jaumann derivatives, and secondly the material derivative is indeed necessary for a Lagrangian frame of reference.*

**231 give a reference to the Equation how to calculate G in Eq. (22) from E and μ.**

*Added an in-line equation and reference*

**Eq 25 why is ρwgwa(t) not included here?**

*We choose to apply ice shelf stresses rather than ocean pressure, which would instead be appropriate for a calving front (for the Rutford ice stream whose geometry we approximately mimic the GL is very far from the Ronne calving front.*

**Fig 3 It might be important to highlight that the grounding line is not just upstream, but also sideways of the model.**

*Done*

**237 Which physical process generates this buttressing stress?**

*This term is added to simulate the buttressing downstream of the model domain, caused by lateral stresses not included in the downstream BC. Although it is not necessary, it allows for a relatively uniform ice shelf velocity in the domain (as is observed on the Rutford Ice Stream).*

**239 and Eqs. (25) and (26): use consistently ρi , or drop the index "i" consistently.**

*Subscript i is not used at any point for ice density, we use ρ without a subscript for ice density, to avoid confusion when using index notation (i.e. Eq. 19)*

**240 I have no idea what an "elastic foundation" is. It looks like you just apply a normal stress on all faces in contact with the ocean, or at least this is what it should be. "Elastic" makes no sense here.**

*Reworded this to avoid confusion. The BC is implemented as an elastic spring foundation and there is an exact mathematical equivalence between ocean pressure and this elastic foundation formulation (e.g. see Gudmundsson 2011).*

**243 If this corresponds to the green "till" area in Figure 3, please say so.**

*Done*

**246 How is this boundary condition implemented? Is a Dirichlet (i.e. velocity) condition prescribed, the magnitude of which is calculated from the basal stress of the last time step solution?**

*The Weertman style sliding law is implemented by including deformable elements beneath the ice, whose rheology simulates the power sliding law. As such, the surface deformation of these sub-ice elements mimics the basal motion, for a given basal traction. This enables us to solve for basal ice velocity implicitly within the same time step and with the Newton-Raphson method.*

**256 simply call this section "Discretization"**

*Done*

**259 Leave away this sentence, we typically also use HEX20 or even better HEX27, so this is standard.**

*To our knowledge this type of element is not at all standard in glaciology and so a description is necessary.*

**260 The model mesh appears extremely coarse for the task at hand. Especially the horizontal discretization in a shear zone should be considerably smaller to resolve the stress concentration there, and the element layer on the sides should be bigger than 2 elements. Or at least some model experiments with double mesh density should be used to show that the chosen resolution is sufficient to resolve the relevant spatial scales. Why was a mesh with such a complicated structure chosen for a block, where a structured mesh, possibly refined in the areas of interest, would have sufficed?**

*We ran the n3xyz simulation (i.e. the one in which bending stresses generate the largest Msf signal) with double the horizontal mesh resolution and analysed the results. The maximum difference in the Msf amplitude between the two simulations was 3% in the ice shelf shear zone. The maximum difference in ice velocity was 2.5% at the ice shelf front. Apart from these very slight changes in amplitude, the model results were identical. Model run time with this setup increased by an order of magnitude. We have added a comment on this in the new manuscript.*
*We started with a structured mesh, with a 90 degree angle in the grounding line where the ice ungrounds next the shear margin, but this unnatural grounding line produced larger stresses and we opted to give the mesh a more natural curved grounding line. As a result of this, it became convenient to use an unstructured mesh, however, the elements are only slightly distorted as compared with the structured mesh we began with.*

**260 Is the mesh moving? If not, how big are the errors in transient stresses?**

*The mesh is moving*

**267 ff.: better explain what n1, n3 means, and leave away the "denoted n3xyz" and similar.**

*Done*

**268 space missing in x, y, z.**

*Fixed*

**Fig 4 It is somewhat confusing to have the horizontal axis reversed as compared to Figures 1 and 3. Better mirror those. Also indicate what is shown with these contours. Are these velocity amplitudes? Or vertical displacement amplitudes at the surface, or the base of the ice? Or horizontal displacements? Additionally, it would help saying that these are map-plane views (I think), of maybe the surface, or average, quantities. Instead of n=1, n=3, the panels should be labeled with the codes of the model experiments e.g. n1xyz. Velocity contours could also be shown for lower velocities, e.g. 0.1, 0.5, 1., 2., such that panels b and c show something interesting**

*All the above has been implemented into new versions of the figures and figure captions.*

**288 now this mysterious term is called the "ocean foundation BC", where it was "elastic" in line 240. Just call this the normal stress. Also parenthesis is missing.**

*Fixed*

**291 So why FRIS, if everything else, including GPS, is from RIS?**

*Changed to Ronne Ice Shelf*

**294 Don't call this "full-Stokes". Either you solve the Stokes equations, or you don't. But here you claim to solve with a viscoelastic rheology, which is again something quite different.**

*See our reply to the main comment*

**295 see 294, and please don't perpetuate this stupid "full-Stokes" thing.**

*See our reply to the main comment*

**297 Please specify which displacements these are: horizontal or vertical? One would expect to see velocities, given the theoretical section and the model description.**

*Specified that these displacements are horizontal. We show displacements rather than velocities because, since these are easiest to observe, almost all previous literature discussed tidal effects on ice flow in terms of displacements.*

**301 "medial line": better write "ice shelf center line"**

*We feel that medial line is more concise and better gets across that this is a line of symmetry through the entire model.*

**308 better "For experiment n1xyz …"**

*Done*

**323 This scaling makes the comparison with Figure 4 difficult. At least you should indicate the grounded portion on the horizontal scale, this might be estimated to be 0.1? Or even less, since the crosses are the stresses extracted at element centers, or element face centers (actually maybe integration points, unless projected to the nodes)?**

*The scaling helps make a comparison to the results of section 3. The grounded sidewall is not shown in either figure 4 or 5 (W is defined as the ice shelf half width, not the half width of the entire domain).*

**325 Since you have symbols, please use those in the descriptions.**

*Done*

**327 units of wa = 2 m. But this was already said in line 321.**

*Added units*

**328 Don't call yy "longitudinal", this is plain confusing. It is the cross-flow component.**

*Done*

**329 where is $\pi/4\lambda$ in the figure? Please help the reader in the figure, or improve the description.**

*The flexural wavelength lambda, as would be calculated by elastic beam theory (Eq. 6), cannot be directly compared with our model. We could estimate the model lambda from the displacement or stress curves but this does not seem like a useful exercise. Removed the pi/4lambda comment to avoid unnecessary complication.*

**333 Probably first tell the reader that τxy is the boring component holding back the weight of the ice, that you would get irrespective of rheology.**

*This is pointed out earlier in the paper*

**333 Also show, and discuss, τE, this is the main point of the paper.**

*Effective stress added to Fig. 5 and new Fig. 6, with discussions added.*

**335 again, no full-Stokes. And as said just above, this part is boring since you get that with any rheology, it's just force balance.**

*As we state at its start, the point of this paragraph is to check some of the assumptions in section 3, one of which was that tau_xy varies linearly with distance from the sidewall.*

**Fig 5 Indicate the grounded part with a colored bar along the horizontal axis, or a vertical dotted line, or similar. Also plot the effective stress, since this is the quantity that is crucial for the whole argument.**

*None of this figure shows results on grounded ice. Effective stress has been added*

**334 This section should go to the discussion.**

*In principal we agree but this is short and flows nicely where it is, whereas it's difficult to see where to put it into the discussion where it doesn't break up the flow.*

**336 Why is τxx increasing downstream? Is this already the effect of the end of the domain, or the standard ice shelf extensional stress?**

*These are standard ice shelf extensional stresses, which begin to dominate away from the sidewall, added a comment on this.*

**340 again, why full-Stokes? This is visco-elastic!**

*See our reply to the main comment*

**348 No, you have not shown that, at least not in the paper. Why aren't any plots of the velocity-spectrum, together with the forcing, shown. Or at least curves of the time-variation of forcing and response. After reading on I found Figure 6 which sort of shows this. So why is this not presented in the "Results" section, such that it could be meaningfully discussed?**

*The amplification of the Msf signal downstream of ice stream GLs is shown in the results section in figure 4a and this point is made in the paragraph that presents these results at the start of the section.*

**355 I sincerely doubt this claim when using a spatial resolution of only 300 m. This effect might be quite important, but will be smoothed out by the horizontal and vertical approximation functions.**

*Quadratic elements can capture this across-flow velocity profile very nicely, tests with double mesh resolution produced a very similar number.*

**362 This period-doubling has been nicely shown in the theoretical part, although with some doubts concerning the problem with viscous/elastic stresses. Now one would like to see this frequency-doubling also in the FE model results. So it would be very nice to have a plot with forcing and response at different points on the domain. Maybe the argument is that the stresses are highest at high tide, and therefore viscous deformation rates. But since everything is transient, and you have a viscoelastic membrane that is bent up and down, there must be location dependent delays.**

*This figure has been added*

**362 Why at high and low tide? Vertical velocities are highest during rising and lowering. You could/should investigate this claim by just extracting the second invariant from your transient model runs.**

*This is now shown in figure 6 and the phasing discussed*

**365 "medial line"**

*See earlier response*

**Fig 6 It is not clear what is shown in this figure. Is this the FFT (frequency spectrum) of the model response? What was the strength of the forcing? It would be very helpful to also show the time series. Panel (b) should be labeled with (m/d), as it shows velocities. Panel (c) (missing) should show the effective stress in the shear margin. This figure would also be a good place to compare the results from the three model runs, maybe as colored bars.**

*The figure is now introduced more clearly, explaining exactly what it shows. The suggestion for panel c has been included in a new figure.*

**371 "…than the run time of 60 days".**

*Changed to "are not resolvable with a simulation time of sixty days"*

**375 I think this is "vice versa" (good old Latin).**

*Fixed*

**402 Again, no "full-Stokes".**

*See our reply to the main comment*

**414 Add some discussion of additional, state-changing processes here (see general comment).**

*Added a new paragraph discussing these processes*

Reply to Victor Tsai

**Major Comments:**

**In reality, the grounding line does not act like a fulcrum and is not fixed. Although the authors have discussed the possibility of grounding line migration somewhat, they have not discussed whether the bending stresses simulated near the grounding line might be overestimated because of the lack of migration (which alleviates the need of the grounding line to bend somewhat). Because the grounding line is assumed to be pinned ("clamped"), they cannot evaluate the possibility that asymmetries in grounding line migration may produce a strongly nonlinear ice shelf flow response (as in Robel et al. 2017, see later comment). This fixed nature of the assumed grounding line therefore seems to be a very important difference between the simulation result with reality, and must be discussed. At a minimum, the authors should describe why they expect their modeling framework to still be useful despite the simplifications.**

*This point raises two separate issues: overestimating the magnitude of bending stresses and the potential role of GL asymmetry to generate an Msf signal, and so we address each in turn. Firstly, regarding bending stresses, although it is not mentioned in the manuscript we tested whether allowing the GL to migrate (for a steep bed slope) had a major impact on the magnitude of bending stresses and this was not the case. For a positive vertical tidal motion of 2m, effective stress at the pinned GL node was 67% greater than for the migrating case, however this large stress is highly localised and the depth averaged effective stress at the grounding line is only 12% greater for the pinned GL. The differences in stress will have some impact on the strength of the mechanism described, but we do not think this error to be any worse than other simplifications such as an isotropic rheology and lack of damage in the shear margin. We have added a discussion on this issue in the revised manuscript. On the second point, as we mention in the manuscript, we intentionally do not allow the GL to migrate in order to isolate the nonlinear rheological mechanism that we are proposing from this alternative mechanism. Since there is currently no strong evidence of GL migration in this area and bed slopes around the GL are not known, it would be difficult to properly model this effect anyway. Also, GL asymmetry was first proposed (and modelled in some detail) as a mechanism a few years previously (see section 3.2 of Rosier et al., 2014).*

**2. It is a basic mathematical fact that a nonlinear process forced at more than one frequency will produce a response at harmonics and beats of those frequencies. The authors claim later in the paper that the flexure mechanism is the only way to produce the M4 response, but they have not proven that other nonlinear processes could not produce such a response. Indeed, Robel et al. 2017 makes this exact point in their equations 11-13. Which brings up the next point. . .**

*We absolutely agree with the point made here, certainly any nonlinear process will produce other frequencies as we discuss in the paper. This needed to be made clearer in our manuscript and we are not trying to claim that this mechanism is the only one capable of producing these high frequencies. Our argument is that there should be a large difference in the amplitude of the response in ice velocity at these higher frequencies that could help diagnose which mechanism is at play. In the Robel (2017) mechanism, the primary response over one tidal cycle is to increase velocity at high tide and decrease velocity at low tide. As the reviewer points out, other frequencies will be in the velocity waveform because the response is nonlinear. However, in the mechanism we put forward, the primary response over one tidal cycle will be to increase velocity twice during one tidal cycle (i.e. precisely at the higher frequencies), and so the high frequencies can be expected to be of much larger*

*amplitude. In this way, observations of a strong velocity response at these frequencies would be evidence that this mechanism is playing an important role. The main point we are raising, therefore, is that while all non-linear processes can give rise to M4, MS4, Msf etc. the ratios of those amplitudes will, in general, be different. Now that we have seen the alternative mechanism put forward by Robel et al. 2017 we can compare the relative strength of the frequencies generated and this reveals that the dominant frequencies generated in ice shelf velocities are M2 and S2. Since our analysis of RIS ice shelf velocities finds the MS4/M4 frequencies dominate, as in our mechanism, we think this is compelling evidence that the flexural ice-softening mechanism is primarily responsible for the observed Msf response on Rutford. We do realize that in our original manuscript much of this was not particularly well articulated, and we have largely re-written this section in order to get our point across more clearly.*

**3. There needs to be much more engagement throughout this paper with the arguments put forward by Robel et al. 2017. While we recognize that this paper was published near the time of submission of the current manuscript, the fact that the article discusses so many of the same issues, including many of the main points of the present manuscript, while also proposing a different basic mechanism related to asymmetries in contact stress from asymmetric grounding line migration, obliges the authors to discuss the Robel et al. paper and contrast their work with that work. For example, at a number of points, it is claimed that the tidal flexure mechanism is the only way to produce an increasing Msf signal in the shelf, which is also what Robel et al. 2017 claims, and the authors also claim that previous models do not reproduce observations in floating ice shelves (which is not true anymore due to the Robel work). Lines 25, 36, 155-160, 295-300, and all of the discussion and conclusions therefore need modification to be accurate and to appropriately cite the present literature.**

*We have added discussions of the Robel 2017 paper throughout, as is appropriate since we are investigating the same problem but come to very different conclusions (incidentally we did not see a copy of the Robel 2017 paper before submission, as can be easily verified by comparing the dates of submission/publication).*

**4. (Lines 357-361 and elsewhere) What about Msf signals generated in the grounding line and then propagated downstream throughout the shelf? Wasn't this the previous explanation for the ice shelf Msf signal? Something that is not remarked upon in this paper in the temporal phasing of signals, which is important given than the Msf signal appears first in the ice shelf.**

The assumption previously (before the Minchew 2016 satellite observations) has been that the Msf signal was generated upstream of the grounding line (due to a nonlinear sliding law and/or subglacial drainage processes). The Minchew 2016 observations show the phase leading on the ice shelf, and this is replicated in our model. We have added a remark on this point in the discussion. We have also included a figure showing more details of the phasing.

**5. I agree that the elastic response can only ever yield a linear response. However, the elastic response can potentially produce a large signal at the primary tidal frequencies. The authors should at least provide an argument (in the analytic section) as to why the elastic deformation is small and so can be neglected in the analytic section.**

We do not understand the point being made here. We all agree that the elastic part of the Maxwell model can only yield a linear response and we explain in the same paragraph that we are concentrating on the nonlinear response because this is the only thing that can explain the observed Msf signal. We are hence not neglecting the elastic deformation but simply using the fact that the

linear response cannot generate any Msf signal and does therefore not need to be considered in this particular case.

**6. It is clear from the difference between n3xyz and n3xy experiments that confinement plays an important role in producing the Msf signal at an amplitude comparable to that observed at the RIS shelf. What about unconfined shelves? Does this indicate that such shelves should have much less Msf response? What about Bindschadler and the other FRIS ice streams? For example, does this imply that the proposed mechanism does not explain the observations of a significant Msf response at Bindschadler. Also, it would be good to state, early on, that Rutford Ice Stream goes afloat in a trough and remains in that trough, for perhaps ~100 km downstream of the grounding line. A map of ice velocities (like Figure 1b of Minchew 2016) would help put this in context.**

In an unconfined ice shelf (such as an ice tongue) our proposed mechanism would still produce an Msf signal at the main GL, and since there would be no sidewall friction the amplitude of this signal would not decay downstream, unlike in our n3xy simulation. Certainly this mechanism will be strongest for very confined ice shelves of which the outlet of Rutford is a good example but many others exist, for example Evans and Foundation Ice Streams. The Msf response at Bindschadler is far smaller than is observed on the FRIS ice streams (because the semidiurnal tides are of low amplitude) and it seems that this could be easily produced by bending stresses at the GL but it is possible that the pinning point downstream plays a role. Determining this would require more observations, together with accurate measurements of bed slopes and/or migration distances.

**7. One aspect of the Minchew 2016 observations that are not explained by this model is the along-flow variation in strain rate in the ice shelf. That study invokes a possible pinning point to produce such heterogeneity. Perhaps this should at least be remarked upon.**

Our mechanism could in fact produce heterogeneity in any number of ways, through variations in ice properties. The ice rheology in our model is kept intentionally homogeneous to avoid complicating the interpretation and although we use a very simplified Rutford geometry our goal is not to reproduce these observations or indeed discuss them to any great length. That being said, the heterogeneity is interesting and we have added a discussion on this.

**Minor Comments:**

**Line 16-17: Please rewrite for clarity: "the primary ice flow response is at a different frequency than the highest amplitude frequency of tidal forcing"**

*Changed to "on the Rutford Ice Stream (RIS) the primary response is at a fortnightly (Msf) frequency that is not measurable in the vertical tidal motion"*

**Line 26-27: Awkward sentence phrasing**

*Reworded*

**Line 30: rheological behavior and the response to external forcing**

*Added this to the end of the previous sentence*

**Line 79-81: Would be useful if this sentence came much earlier to direct focus to the figure**

*Done*

**Line 122: There are not separate viscous and elastic stresses. There is simply one stress which causes both viscous and elastic deformation. This sentence therefore should be rewritten to clarify.**

*Reworded this sentence to clarify*

**Equation 10: This is clearly only do-able in this way for n=3. Could you use a perturbation approach (or expand about \tau_xy) to solve for general n?**

*This would be an interesting future extension to the work, as we mention at the end of the paper it could be possible to infer a value for 'n' from observations of this nonlinearity and so understanding how different values of 'n' produce different frequencies would be important. For now, since 'n=3' is a very standard assumption in glaciology, we prefer this simpler approach since it only produces one set of frequencies and is easier to interpret.*

**Line 176: How large are the linear elastic and damming stresses in comparison to the nonlinear viscous changes that you are simulating?**

*In this section we explore the flexural ice-softening mechanism in isolation, to simplify the analysis, and do not include these terms. Both of these effects are included in our 3D model, and we have added a comment pointing this out in this sentence.*

**Line 231: It needs to be explained more clearly why the elastic modulus in the analytic calculation has to be so different from the numerical calculation. (Why does this produce such high bending stresses in the analytic case?)**

*We have reworded this paragraph to explain this better.*

**Line 240: What is an elastic foundation? Perhaps use a clearer term.**

*Changed to normal stress*

**Line 245-250: Should say why you can be sure that the nonlinear sliding law isn't producing the Msf response. (I can see that it isn't because all the numerical experiments use the same sliding law.) Also, the authors should comment about what this implies about the results in Gudmundsson 2007, 2011 and other studies. . .**

*We have added several sentences in the discussion which cover this*

**Figure 4: Msf amplitude in what quantity? (same comments for line 298)**

*Expanded on this description (Msf amplitude in horizontal surface ice displacements)*

**Line 303-308: Should state outright here that the amplitude of Msf response is an order of magnitude less than in the n3xyz experiment.**

*In fact the Msf amplitude at the main GL is more or less the same as the n3xyz experiment (since bending stresses are generated here), we have changed the wording of this paragraph to clarify how the Msf amplitude compares to the n3xyz experiment.*

**Line 327: units of w_a?**

*Added units (m)*

**Line 367: Would the response at M2 and S2 frequencies be larger if the elastic modulus was different? The relative difference between linear elastic and nonlinear viscous responses will be a strong function of the relative size of viscosity and elastic modulus.**

*The linear elastic response, which will largely consist of M2 and S2 frequencies, would undoubtedly be larger if the elastic modulus was different. However, the elastic modulus is not a tuneable parameter, its value is set in order to match a Burgers rheology at tidal frequencies, as in Gudmundsson 2011.*

**Line 408: Remove "to this day"**

*Done*

**At the end of this file we have included a marked-up manuscript showing changes to the paper. Please note that some reviewer comments were implemented to the originally submitted .tex file before a new version was created and so not every single change is highlighted in red, however the majority are shown. Our apologies for this oversight, we hope that the marked-up version can still serve as a useful guide to highlight the main changes.**

[revised manuscript text omitted]

---

## Author Response (AR2)

Dear Olivier,

Thank you very much for considering our manuscript for publication in 'The Cryosphere' and for your helpful comments. Please find below our replies (in italics) to each of your points (in bold).

Kind regards,

Sebastian Rosier and Hilmar Gudmundsson
* * *
**- line 80: flexural ice- softening -> flexural ice-softening**

*Done*

**- line 90: what is h in Eq. (4)-(6) should be specified as from Fig. 1 it seems to be the ice thickness at the front, whereas in the analytical solution it is the mean(?) or local(?) thickness. Also, it is mentioned line 113 that this solution is derived for an infinitively long ice-shelf (which I guess means with periodic boundary conditions in the x direction), which seems in contradiction with a constant thickness gradient along x? How the assumption that h is not constant departs from this assumption should be discussed (or at least clearly mentioned).**

*Ice thickness (h) is now defined in the figure caption and the relevant portion of text.*

*Ice thickness 'h' in our analytical solution refers to local ice thickness and a thickness gradient exists which contributes to the driving stress 'F_d'. Our analytical solution provides u(y) at some point downstream of the grounding line for a particular ice thickness at that point and the magnitude of bending stresses varies with the ice thickness.*

*The beam equations we use are 1-dimensional and in the past have been used in the context of a floating ice tongue protruding outwards perpendicular from the coast. We focus on the across-channel bending stresses, so our wording is admittedly poor because in this case the equations are only strictly correct for an infinitely wide ice shelf. This is not due to a periodic BC but due to other BC's which must be satisfied and are outlined in Appendix B. In our geometry, ice thickness is a function of x but not y and so, as long as we focus on a region sufficiently far from the GL that 2-D effects are negligible, the thickness gradient in x does not matter. In Appendix B we demonstrate that, although our geometry is not infinitely wide and so the BCs are not the same, the equations are still a good approximation for a sufficiently wide ice shelf such as is found at the outlet of Rutford Ice Stream. So it is only the beam equations (from which we obtain bending stresses) which were derived for an infinitely long ice shelf, we show that they still work for our geometry and do not make this infinitely long assumption anywhere in our analytical solution. We have reworded things in a few places to hopefully make all of this clearer.*

**- line 92: the GL that the -> the GL so that the(?)**

*Changed to "the GL such that the…"*

**- line 122: w_a is noted W_a in Fig. 1 (make it consistent). The Poisson's ratio is usually \nu?**

*Fixed w_a and replaced all instances of \mu with \nu for Poisson's ratio.*

**- line 143: s is the surface elevation, b is the bed elevation**

*Done*

**- Equation 18: why not u_shift instead of n_shift (which looks like a ratio of velocity, not a velocity)?**

*We prefer to continue calling this "n_shift" because it emphasises that it is the nonlinearity arising from "n" that contributes to the increase in velocity due to tidal bending.*

**- Caption Fig. 2: in table 3 -> in Table 3**

*Done*

**- line 198: Then I don't understand because it is mentioned above that only the non linear viscous term is conserved and that elasticity is neglected? Did I miss something? There is no E in Eq. (18)? Which value for A has been used? It should be mentioned in Table 1.**

*In order to calculate the stresses generated by tidal bending we use elastic beam theory, these stresses are then used in the purely viscous analytical solution which is possible because of the Maxwell rheological model that we use, in which elastic and viscous stresses are equal. Thus, E enters in Eqs. 4, 5 and 6 which define the stresses. The wording is probably confusing though and so we have changed it to make this clearer.*

*The value used for A is now mentioned in Table 1.*

**- line 248: \rho_i should be \rho**

*Done*

**- line 250: similarly to reviewer #1, I don't understand what is an elastic foundation. Are you just applying a normal stress equals to the water pressure on that surface? If yes, why do you need to introduce this concept of "elastic foundation"? Or is it more complex, like the "viscous spring" term in Eq. (15) of Durand et al. (2009)? Then it should be mentioned.**

*The elastic foundation BC that we implement in our FE software is mathematically exactly equivalent to applying a normal stress to the face of the element. We choose to apply the ocean pressure in this way because it requires less iterations to converge than directly applying a normal stress, by reformulating the force to be a function of the unknown ice shelf base position. No additional dampening terms (as in Durand et al. 2009) are needed to aid stability. We reworded this slightly and added a citation to Gudmundsson (2011), where this is described in more detail.*

**- Figure 3: margin till -> margin BC?; ice stream till -> ice stream BC?**

*Done*

**- line 330: no space after MS_4**

*Done*